# Arbitrary-Order Block SignSGD for Memory-Efficient LLM Fine-Tuning

**Yijie Zhou & Shi Pu**[*]
The Chinese University of Hong Kong (Shenzhen)
`yijiezhou@link.cuhk.edu.cn, pushi@cuhk.edu.cn`

## Abstract

We propose **ABSignSGD**, a block-coordinate variant of sign-based descent with flexible block selection that enables memory- and runtime-efficient full-parameter fine-tuning of large language models. We present a unified convergence analysis under mild conditions, covering both the base method and a *majority-vote* extension for distributed training. The latter improves communication efficiency by aggregating only gradient signs rather than averaging full gradients. Experiments on Qwen3-8B, Llama3-8B, and Qwen3-32B, spanning mathematical reasoning and general instruction-following tasks, show that ABSignSGD converges faster per iteration and delivers superior downstream performance while reducing both runtime and memory usage compared to existing methods. Ablation studies further indicate that the memoryless sign-based update naturally complements block-wise updates, explaining the method's strong empirical performance.

## 1 Introduction

Large Language Models (LLMs) achieve state-of-the-art results in reasoning, dialogue, and code generation (Achiam et al., 2023), but specialized applications still require task-specific adaptation (Ding et al., 2023). Fully retraining is prohibitively expensive, making fine-tuning the practical route for domains such as biomedical text (Singhal et al., 2023) or legal text (Chalkidis et al., 2020) and for aligning behavior like multilingual support. Yet even fine-tuning imposes heavy GPU memory demands (Han et al., 2024), motivating methods that reduce memory and runtime without degrading performance. Existing strategies to reduce the memory footprint of LLM fine-tuning span multiple directions. System-level techniques such as quantization (Dettmers et al., 2022; 2023; Lin et al., 2023) modify the numerical representation of model parameters or activations (e.g., storing weights in lower precision), while offloading (Ren et al., 2021; Rajbhandari et al., 2021) changes the storage location of tensors within the hardware memory hierarchy (e.g., moving optimizer states to CPU or NVMe). Complementary to these system-level approaches, algorithmic optimizers form the second major axis of memory-efficient fine-tuning. Among them, zeroth-order methods (Zhang et al., 2024; Liu et al., 2024) represent an important line of work, eliminating backward passes and achieving inference-level memory usage. However, their slow convergence often limits practical applicability for LLM fine-tuning.

In this work, we focus on *first-order algorithmic methods* that aim to reduce memory and runtime without sacrificing performance. Existing methods can be broadly grouped into three families.

**(i) Parameter-efficient fine-tuning (PEFT).** These methods reduce memory usage by training only a small set of additional parameters while keeping the base model frozen. Representative techniques include prefix-tuning (Li & Liang, 2021), prompt-tuning (Lester et al., 2021), and adapter architectures (Houlsby et al., 2019; Pfeiffer et al., 2021). The most widely adopted strategy is Low-Rank Adaptation (LoRA) (Hu et al., 2021), which reparameterizes weight matrices using low-rank factors. PEFT reduces memory and makes fine-tuning feasible on limited hardware, yet usually yields lower performance than full-parameter training.

**(ii) Low-rank projection for full-parameter training.** Methods such as GaLore (Zhao et al., 2024), Fira (Chen et al., 2024), Flora (Hao et al., 2024), and GoLore (He et al., 2024) cut optimizer

---

[*]Corresponding author. The code is available at `https://github.com/yijiezcn/ABSignSGD`.

memory by projecting gradients into a low-rank subspace via SVD or faster/cheaper random projections, sometimes applied intermittently to reduce cost. They preserve full-parameter updates but may face: (a) performance gaps compared to strong baselines like AdamW, (b) incompatibility with standard gradient accumulation under layerwise updates for maximal memory saving, and (c) slow runtime when frequent costly decompositions are needed.

**(iii) Block-wise optimization.** This approach updates only a subset of parameters per iteration. For example, BAdam (Luo et al., 2024) combines block-coordinate updates with Adam, saving memory by storing optimizer states only for the active block and reducing runtime by halting backpropagation at that block. However, Adam's dependence on first- and second-moment estimates conflicts with block switching, requiring frequent state resets that, as shown in our ablations, degrade convergence relative to Adam with full-model updates.

Motivated by these trade-offs, we propose **Arbitrary-order Block SignSGD (ABSignSGD)**, which combines block-coordinate updates with sign-based descent. Leveraging the simplicity of SignSGD, our method offers greater memory savings and extra runtime gains from arbitrary-order updates while maintaining competitive performance. We further introduce **ABSignSGD-MV**, a communication-efficient data-parallel variant that transmits only block gradient signs (1 bit per coordinate) cutting communication cost by $960\times$ (with 30 blocks) over standard DDP (Paszke et al., 2019) without harming convergence. As shown in Table 1, ABSignSGD achieves the lowest memory overhead, smallest communication budget, and fastest runtime.

Table 1: Memory-efficient optimizer comparison. $M$: model parameters (billions); $r$: rank for low-rank methods; $m$: weight matrix dimension (assumed square); $N$: layers (for layer-wise update) or blocks. Memory/communication in GB. ABSignSGD achieves the lowest memory and communication costs and the fastest runtime. See Appendix A for derivations and details.

| Method | Memory Overhead[†] | Comm. Budget[‡] | Gradient Accum. | Runtime Speedup |
|---|---|---|---|---|
| **ABSignSGD** | $\frac{M}{8N}$ | $\frac{M}{8N}$ | ✓ | ✓✓[§] |
| BAdam | $\frac{16M}{N}$ | $\frac{4M}{N}$ | ✓ | ✓ |
| LoRA | $\frac{36Mr}{m}$ | $\frac{8Mr}{m}$ | ✓ | ✗ |
| GaLore | $\frac{8M}{N} + \frac{12Mr}{m} + \frac{4Mr}{mN}$ | – | ✗ | ✗ |
| Apollo | $\frac{8M}{N} + \frac{8Mr}{m} + \frac{4Mr}{mN}$ | – | ✗ | ✗ |

[†] Excludes the $2M$ GB half-precision weights stored by all methods. [‡] For low-rank projection methods, original papers omit communication budgets; sending full gradients costs $4M$ GB—orders of magnitude higher than others—and even low-rank gradients remain comparable to LoRA and far above ABSignSGD. [§] Double checkmark denotes additional runtime speedup from arbitrary-order block updates.

## 1.1 CONTRIBUTIONS

**(i)** We introduce ABSignSGD, a block-coordinate variant of SignSGD that enables *arbitrary-order block updates*, allowing us to tailor the update policy for maximal efficiency (e.g., depth-biased updates; see Contribution (iii)). This design delivers substantial memory and runtime savings while preserving competitive convergence and downstream performance. We further extend the method to distributed training with *ABSignSGD-MV*, which aggregates only gradient signs via majority vote, thereby achieving extreme communication efficiency.

**(ii)** We establish theoretical convergence guarantees under mild assumptions, providing a unified analysis for ABSignSGD and ABSignSGD-MV. Specifically, they achieve $\mathcal{O}(\frac{1}{\sqrt{K}})$ convergence under arbitrary block selection schemes given bounded update intervals.

**(iii)** We introduce a depth-biased update that prioritizes deeper layers, providing runtime speedup without sacrificing performance. Extensive experiments on fine-tuning Qwen3-8B and Llama3-8B for mathematical reasoning and instruction-following show that ABSignSGD achieves the lowest memory footprint, fastest runtime, and superior downstream performance among memory-efficient optimizers. A targeted ablation study further pinpoints the factors driving its effectiveness.

Next, we formalize the fine-tuning setting and present the ABSignSGD algorithm.

## 2 ALGORITHM DESIGN

### 2.1 PROBLEM SETTING

We consider the general unconstrained optimization problem

$$\min_{x \in \mathbb{R}^d} f(x). \tag{1}$$

In the case of LLM fine-tuning, the objective $f(x) = \mathbb{E}_{\xi \sim \mathcal{D}} F(x, \xi)$, where $F$ is the loss function and $\mathcal{D}$ is the data distribution. And one uses a batch gradient $g(x)$ to estimate $\nabla f(x)$.

**Notation**: $[n] := \{1, 2, ..., n\}$ denotes the set of integers from 1 to $n$. Let $\mathcal{P} = \{\pi_1, \ldots, \pi_N\}$ be a partition of $[d]$ into $N$ blocks; that is $\pi_i \cap \pi_j = \varnothing$ for $i \neq j$ and $\bigcup_{i=1}^{N} \pi_i = [d]$. We write $x = (x_{\pi_1}, ..., x_{\pi_N})$, where $x_{\pi_i} \in \mathbb{R}^{d_i}$ collects the coordinates of $x$ indexed by $\pi_i$. For notation convenience, we denote the $i$-th block by $x_\mathbf{i} := x_{\pi_i}$, while $x_j$ denotes the $j$-th coordinate of $x$ for $j \in [d]$. For memory analysis, we assume the model has $M$ billion parameters.

### 2.2 PROPOSED METHOD: ABSIGNSGD

The most widely used optimizer for LLMs is Adam (Kingma & Ba, 2015), which stores first-order and second-order momentum for each parameter. The update rule scales the learning rate adaptively based on these estimates. While effective, maintaining these optimizer states nearly triples the memory required for parameters, posing a significant bottleneck. In contrast, SignSGD (Bernstein et al., 2018) is a stateless optimizer that discards gradient magnitude entirely, relying only on the sign of the gradient for parameter updates:

$$x^{k+1} = x^k - \alpha \cdot \text{sign}(g(x^k)).$$

This memory-efficient approach remains competitive because its dynamics share similarities with Adam (Kunstner et al., 2023; 2024). Sign-based principles have been successfully incorporated into modern optimizers for LLMs, such as Lion (Chen et al., 2023), which leverages the sign of the momentum term. Crucially, recent empirical evaluations (Zhao et al., 2025) show that sign-based methods are comparable to AdamW in both performance and hyperparameter robustness. However, previous applications have focused on full-model updates; combining the efficiency of SignSGD with a block-update framework remains unexplored.

We adopt SignSGD with block-coordinate updates to solve Problem (1). At iteration $k$, the algorithm selects a block $x_{\mathbf{i_k}}$ and updates its coordinates using stochastic gradient signs:

$$x_{\mathbf{i_k}}^{k+1} = x_{\mathbf{i_k}}^k - \alpha \cdot \text{sign}(g_{\mathbf{i_k}}(x^k)), \tag{2}$$

where $g_{\mathbf{i_k}}(x^k)$ is the block gradient estimate. We refer to update (2) as Arbitrary-order Block-Coordinate SignSGD (ABSignSGD). Algorithm 1 presents its complete procedure, along with a communication-efficient variant (see Section 2.2.2). The name reflects its tolerance for flexible block selection: each block only needs to be updated at least once every $B$ steps (see Section 3.1). This property enables customized update rules that speed up training without degrading performance. As one example, we propose a depth-biased selection strategy (see Section 4.1) that updates deeper layers more frequently, yielding additional runtime savings while maintaining strong accuracy.

### 2.2.1 ABSIGNSGD IS MEMORY- AND TIME-EFFICIENT

ABSignSGD offers notable memory savings, requiring only $2M + \frac{M}{8N}$ GB of memory (excluding activations) for training. In contrast, Adam with mixed-precision training requires $18M$ GB of memory. The savings mostly stem from storing optimizer states only for the active block. For comparison, BAdam (Luo et al., 2024), Adam with block updates, consumes $2M + \frac{16M}{N}$ GB memory. ABSignSGD achieves further savings by (i) avoiding moment storage and (ii) only using signs for updates. For an 8B model with $N = 36$, this yields an extra $3.5$ GB memory reduction.

Second, as a block-update method, ABSignSGD also enjoys backpropagation runtime savings when blocks align with neural network layers. As observed in (Luo et al., 2024), computing the gradient for a given layer (block) allows the backward pass to terminate at that layer; updating only the final layer thus eliminates nearly all backpropagation cost. BAdam uses cyclic block updates, yielding about a 50% reduction in backpropagation time. ABSignSGD extends this advantage by allowing *arbitrary* (see Assumption 3.3) block updates, enabling deeper layers to be updated more frequently and further reducing backward-pass computation. In the extreme, updating the last layer for $B - (N - 1)$ consecutive iterations before updating each remaining layer once reduces the average backpropagation time to approximately $\frac{1}{N-1}$ for large $B$. In practice, we develop a strategy (see Section 4.1) that delivers an additional $\approx 20\%$ runtime saving without impairing performance.

---

**Algorithm 1** ABSignSGD and ABSignSGD-MV (local view)

---

**Require:** Initial point $x^0 \in \mathbb{R}^d$; partition $\mathcal{P} = \{\pi_1, ..., \pi_N\}$; stepsize $\alpha$; block-selection rule.
1: **for** $k = 0, 1, 2, \ldots$ **do**
2:      Select block $i_k$
3:      **if** single-agent **then**                                              ▷ **ABSignSGD**
4:          $v \leftarrow \text{sign}\big(g_{\pi_{i_k}}(x^k)\big)$
5:      **else** multi-agent                                          ▷ **ABSignSGD-MV**
6:          $v \leftarrow \text{sign}\Big( \sum_{j=1}^n \text{sign}\big(g_{\pi_{i_k}}^j(x^k)\big)\Big)$       ▷ Aggregate signs with majority vote
7:      **end if**
8:      $x_{\pi_{i_k}}^{k+1} \leftarrow x_{\pi_{i_k}}^k - \alpha \cdot v$
9:      $x_{\pi_i}^{k+1} \leftarrow x_{\pi_i}^k$ for all $i \neq i_k$
10: **end for**

---

### 2.2.2 A COMMUNICATION-EFFICIENT EXTENSION

We further extend ABSignSGD to the data-parallel setting, yielding an extremely communication-efficient multi-agent variant: *Arbitrary-order Block-Coordinate SignSGD with Majority Vote* (ABSignSGD-MV). This method inherits all the memory and runtime benefits of ABSignSGD while substantially reducing inter-agent communication.

In ABSignSGD-MV, $n$ agents compute stochastic gradients in parallel. At iteration $k$, all agents update the same block $x_{\mathbf{i_k}}$ according to

$$x_{\mathbf{i_k}}^{k+1} = x_{\mathbf{i_k}}^k - \alpha \cdot \text{sign}\left( \sum_{j=1}^n \text{sign}\left(g_{\mathbf{i_k}}^j(x^k)\right)\right), \tag{3}$$

where $g_{\mathbf{i_k}}^j(x^k)$ denotes the block stochastic gradient computed by agent $j$. Unlike the standard approach, which applies $\text{sign}(\sum_{j=1}^n g_{\mathbf{i_k}}^j(x^k))$, ABSignSGD-MV first takes the sign of each agent's block gradient, then aggregates these signs via majority vote.

With this design, each iteration requires agents to exchange only the *signs* of the block gradient, amounting to just 1 bit per coordinate, rather than full-precision values (32 bits per coordinate). For $N = 30$ blocks, this reduces communication volume by $960\times$ relative to the standard `PyTorch DistributedDataParallel` implementation (Paszke et al., 2019). Under the same setting, the reduction is $32\times$ compared to BAdam, and $4.5\times$ compared to LoRA with rank $r = 8$ and internal dimension $m = 4096$. Moreover, the MV estimator is asymptotically more robust under heavy-tailed noise, as indicated by Theorem 3.5.

### 2.2.3 PRACTICAL CONSIDERATIONS: COMPATIBILITY AND LIMITATIONS

ABSignSGD inherently applies extreme gradient quantization by reducing each gradient coordinate to a single bit, making any additional gradient quantization redundant. However, *weight quantization* and *activation quantization* remain fully compatible and can be combined with ABSignSGD to further reduce memory footprint and runtime. Similarly, *offloading* of weights or activations is compatible, though optimizer-state offloading is irrelevant due to ABSignSGD's statelessness.

Convergence of sign-based updates requires the sign-agreement probability to exceed $0.5$. Prior work Safaryan & Richtárik (2021) shows divergence on toy problems when this condition fails, so such methods may underperform under certain regimes, e.g. with extremely small batch sizes. Appendix G.1 shows that ABSignSGD is indeed more sensitive to noise (i.e., decreasing batch sizes) than the baseline. Crucially, however, it avoids breakdown and maintains a faster convergence rate than BAdam even under the extreme noise of batch size 4. Furthermore, in our main experiments (Section 4) with a relatively small batch size of 16, ABSignSGD demonstrates a substantial performance lead. The observed sensitivity can be partially attributed to the absense of orthogonal mechanisms (e.g., momentum and adaptive learning rate) rather than the sign update itself. A promising future direction is to incorporate such stateful techniques via system-level offloading. Since block-coordinate methods require only the active block's state at each iteration, the I/O bandwidth demand is minimal. This allows for the offloading of optimizer states to enable momentum-based variance reduction without compromising the method's ultra-low memory footprint or runtime efficiency.

## 3 THEORETICAL ANALYSIS

Having introduced the algorithm and its distributed extension, we now present a theoretical analysis, establishing convergence guarantees for both under mild conditions within a unified framework

### 3.1 ASSUMPTIONS

**Assumption 3.1** (*L*-smoothness and Lower Boundedness)**.** *The function $f$ is L-smooth and lower bounded, i.e. $\|\nabla f(x) - \nabla f(y)\|_2 \le L\|x - y\|_2$ and $f(x) \ge f^*, \forall x, y$.*

**Assumption 3.2.** *For each element in the gradient estimator $g(x)$ , its sign aligns with that of the ground truth gradient $\nabla f(x)$ with a probability larger than 1/2. Namely,*

$$\rho_i(x) = \mathbb{P}[sign(g_i(x)) = sign(\nabla_i f(x))] > \frac{1}{2},$$

$\forall x \in \mathbb{R}^d$ *and all $i \in [d]$.*

Assumption 3.2, also referred to as the Success Probability Bound (*SPB*) in (Safaryan & Richtárik, 2021)(equivalently, the *sign-agreement probability* bound) has several sufficient conditions. One such condition holds when the gradient noise is unimodal and symmetric, a property observed in many deep learning tasks (Bernstein et al., 2018). Another guarantee arises if the gradient noise variance satisfies the element-wise bound $\sigma_i^2(x) \le c_i\, g_i^2(x)$ and the mini-batch size exceeds $2\max_i c_i$.

**Assumption 3.3** (Bounded Update Interval)**.** *There exists a positive integer $B$ such that, for every $t$, each block index $j \in [N]$ is selected at least once within the interval $\{t, \ldots, t+B-1\}$. Equivalently, for every $t$ and every $j \in [N]$, there exists some $k \in \{t, \ldots, t + B - 1\}$ with $i_k = j$.*

This assumption affords the algorithm substantial flexibility in block selection, which, as analyzed in Section 2.2.1 and corroborated by later results, can further reduce runtime.

### 3.2 CONVERGENCE RESULTS

We establish convergence theorems for both ABSignSGD and its Majority Vote variant under this customized alignment norm, following the formulation in (Safaryan & Richtárik, 2021).

**Definition 3.1** (Alignment norm)**.** *Let $g(x) \in \mathbb{R}^d$ and $\{w_i(x)\}_{i=1}^d$ be alignment weights with $0 \le w_i(x) \le 1$. Define the alignment norm*

$$\|g(x)\|_{\mathcal{N}} := \sum_{i=1}^d w_i(x)\, |g_i(x)|.$$

*For **ABSignSGD**, we use $w_i(x) = 2\rho_i(x) - 1$, where $\rho_i(x)$ is from Assumption 3.2. For **ABSignSGD-MV**, $w_i(x) = 2I(\rho_i(x); l, l) - 1$, where $l = \lceil (n + 1)/2 \rceil$ and $I(\cdot; \cdot, \cdot)$ is the regularized incomplete beta function.*

Although termed a "norm," $\|\cdot\|_{\mathcal{N}}$ is not a true mathematical norm; it is a weighted $\ell_1$-type measure in which each coordinate's contribution is scaled by its likelihood of sign agreement with the true

gradient. Coordinates with higher $\rho_i(x)$ receive greater weight, reflecting their higher expected contribution to descent and their importance for convergence guarantees. In the single-agent case ($n = 1$), the ABSignSGD-MV weights reduce to those of ABSignSGD.

We now present a unified convergence guarantee for both ABSignSGD and its Majority Vote (MV) variant under this alignment-norm framework. The result applies to both single- and multi-agent settings, differing only in the definition of the alignment weights.

**Theorem 3.4** (Unified Convergence of ABSignSGD and ABSignSGD-MV). *Given Assumption 3.1 to 3.3 and assuming identical block size, ABSignSGD and ABSignSGD-MV converge as follows:*

$$\frac{\sum_{k=0}^{K-1} \mathbb{E}\|\nabla f(x^{kB})\|_{\mathcal{N}}}{K} \leq \frac{f(x^0) - f^*}{\alpha K} + \alpha L d \big( B \big(1 + \frac{1}{2N}\big) - \frac{N+1}{2} \big)),$$

*with different weights $w_i$ as defined in Definition 3.1.*

When the smoothness and lower-boundedness, *SPB*, and bounded-update interval conditions hold, and block sizes are identical, both ABSignSGD and its Majority Vote (MV) variant converge up to a steady-state term determined by the step size, the block-update interval, and the number of blocks, with convergence defined in the alignment norm. The single- and multi-agent settings are unified within a common proof framework, differing only in the definition of the alignment weights. With a proper step size $\alpha = 1/\sqrt{K}$, we arrive at a $\mathcal{O}(1/\sqrt{K})$ convergence rate, which persists despite aggressive reductions in memory, runtime, and, via MV, communication cost.

Although MV discards gradient magnitude information, it can match or even outperform gradient averaging in certain regimes, particularly under heavy-tailed noise (a common phenomenon in deep learning (Gurbuzbalaban et al., 2021)). This robustness stems from its resistance to confidently misaligned outliers, as it ignores gradient magnitudes and thus avoids amplifying their influence. This is validated by the following result.

**Theorem 3.5** (Informal Statement of Theorem C.3). *Under heavy-tailed noise, Majority Vote is asymptotically a superior sign estimator compared to aggregation by arithmetic mean.*

Together, these results establish that ABSignSGD and its MV variant retain strong convergence guarantees while offering robustness under heavy-tailed noise, providing the theoretical foundation for the empirical studies in Section 4.

## 4 EXPERIMENTS

We now evaluate ABSignSGD against leading memory-efficient fine-tuning optimizers, measuring memory usage, runtime, convergence speed, and downstream performance.

### 4.1 EXPERIMENTAL SETUP

**Tasks and datasets.** We fine-tune QWEN3-8B on OpenMathInstruct-2 (Toshniwal et al., 2024) (50K samples) for mathematical reasoning and Stanford–Alpaca (Taori et al., 2023) (35K samples) for general instruction following. The 8B scale is the primary focus, as it represents the most common use case for full-parameter fine-tuning. Each dataset is fine-tuned separately and evaluated on task-specific benchmarks: `math-evaluation-harness` (Gou & Zhang, 2025) for math reasoning, and MT-Bench with a GPT-5 judge via `FastChat` (Zheng et al., 2023) for instruction following, following the official protocol. Results on LLAMA3-8B show similar trends and are reported in Appendix E. Scalability is further validated by fine-tuning QWEN3-32B on the math task, where consistent performance gains are observed; these results are provided in Appendix E.

**Baselines.** We compare ABSIGNSGD with leading memory-efficient optimizers from three families: low-rank adaptation (LORA), low-rank projection (GALORE, APOLLO), and block-coordinate (BADAM) methods. To narrow the comparison, we exclude any method that inherently uses orthogonal techniques, like quantization/offloading, as part of its core design, such as QLoRA (Dettmers et al., 2023). We also exclude sign methods that rely primarily on momentum buffers, such as Lion (Chen et al., 2023), since these incur substantially higher memory costs unless paired with orthogonal techniques. Hyperparameters follow each method's official implementation where possible, with all other settings matched. We adopt gradient checkpointing while disabling gradient

accumulation (even when possible) to avoid runtime bias. No offloading, quantization, or other memory-saving techniques are applied. Further details are deferred to Appendix D.

**Block partitioning and selection strategy.** The model is partitioned into $N$ blocks at the layer level, where each Transformer layer (including attention and FFN modules) constitutes a single block (resulting in $N = 36$ for Qwen3-8B). Building on this layer-wise partition, our framework supports arbitrary block selection. As discussed in Section 2.2.1, prioritizing deeper layers yields measurable runtime gains. We also hypothesize that this may help mitigate catastrophic forgetting, as shallower layers tend to encode more general features (Howard & Ruder, 2018). We leave a thorough empirical validation of this effect to future work. Moreover, we seek to avoid repeatedly updating the same block in succession, which risks premature convergence to poor local minima. To balance these considerations, we adopt an event-driven *depth-biased* update rule.

To implement this strategy while satisfying the bounded update interval assumption (Assumption 3.3), an event-driven update rule is adopted. Each block $i$ is assigned a fixed "virtual update cost" $\tau_i$, which serves as a hyperparameter to control the relative update frequency. A "next-ready" virtual timestamp $T_i$ is maintained for each block, initialized to $\tau_i$. At each iteration $k$, the algorithm selects the block with the minimum timestamp, $i_k = \arg\min_i T_i$, performs the update, and increments the timestamp: $T_{i_k} \leftarrow T_{i_k} + \tau_{i_k}$. This scheme ensures that every block is updated at least once within a fixed interval $B$ (the derivation of $B$ and a concrete execution trace are provided in Appendix D.2). The costs are defined as $\tau_i = N + c(N - i + 1)$, where $i$ is the block index (1 being shallowest) and $c$ is a bias coefficient. In the experiments, $c = 10$ is used, ensuring deeper blocks are updated more frequently. Note that ABSignSGD and its MV variant remain fully synchronous.

## 4.2 MEMORY FOOTPRINT AND RUNTIME

Table 2 reports the peak reserved GPU memory during training and the wall-clock runtime. AB-SignSGD attains the smallest peak memory usage at 20.29 GB, about 2 GB lower than LoRA and Apollo, and nearly 3 GB lower than BAdam and GaLore. In terms of runtime, ABSignSGD is ≈20% faster than BAdam and roughly twice as fast as LoRA, with even larger gains over projection-based methods. The speedup relative to BAdam indicates that the *depth-biased* block-selection scheme delivers additional runtime benefits beyond those of block updates alone. Overall, the empirical evidence reinforces ABSignSGD's suitability for large-model fine-tuning under tight memory budgets.

Table 2: Maximum reserved GPU memory and runtime for fine-tuning Qwen3-8B on 50K OpenMathInstruct-2 samples for 3 epochs. ABSignSGD achieves the lowest memory footprint and fastest runtime among all compared methods.

| Metric | ABSignSGD | LoRA | GaLore | BAdam | Apollo |
|---|---|---|---|---|---|
| Mem. Reserved (GB) | **20.29** | 22.54 | 23.47 | 23.19 | 22.58 |
| Runtime (h) | **2.66** | 5.51 | 12.77 | 3.32 | 6.64 |

## 4.3 CONVERGENCE SPEED

We present convergence curves for fine-tuning Qwen3-8B on the OpenMathInstruct-2 dataset, using each method's optimal step size determined via grid search (details in Appendix D). As shown in Figure 1, ABSignSGD reduces training loss more quickly than all baselines, both in terms of training token count and wall-clock time.

This improvement arises from combining sign-based updates, which deliver strong per-iteration progress, with a depth-biased update rule that reduces runtime. Together, these features allow AB-SignSGD to achieve lower loss in fewer updates and finish training sooner, making it well-suited when both convergence speed and runtime are critical.

The distributed Majority Vote (MV) variant is further analyzed in Figure 1. Although not emphasized in the caption, Figure 1 (Left) includes the convergence curve for ABSignSGD-MV running on four agents with an identical global batch size to the single-node version. The MV variant tracks the non-MV baseline closely, confirming that the sign-based aggregation preserves convergence speed.

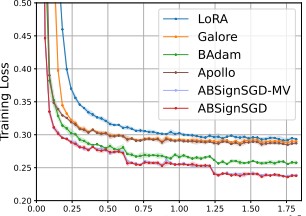 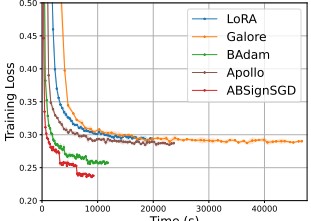 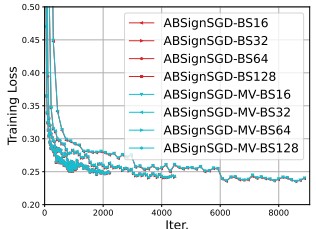

Figure 1: Training loss for Qwen3-8B on OpenMathInstruct-2. **Left** & **Middle**: Convergence comparison against baselines in terms of tokens and wall-clock time (ABSignSGD-MV is excluded for time comparison as it uses multiple agents). ABSignSGD achieves the fastest reduction in both metrics. **Right**: Robustness of ABSignSGD-MV. The curves show loss vs. iterations as the number of agents increases from 1 to 32 (fixing local batch size). The MV variant closely tracks the single-agent baseline regardless of the agent count, demonstrating high scalability.

Figure 1 (Right) further highlights the method's robustness to scaling. With the local batch size fixed at 4, the number of agents is increased from 1 to 32 (scaling the global batch size from 4 to 128). ABSignSGD-MV consistently tracks the single-agent baseline across all settings, confirming the predictions of Theorem 3.5 and demonstrating that the method remains stable even as the number of voting agents increases. Further results on a fixed global batch size is deferred to Appendix G.2.

## 4.4 DOWNSTREAM TASKS PERFORMANCE

To assess the practical impact of faster convergence, we evaluated the fine-tuned models on downstream tasks. The results show that ABSignSGD's optimization efficiency translates into stronger generalization across both specialized and general domains.

**Mathematical Reasoning.** On diverse mathematical benchmarks, Qwen3-8B fine-tuned with ABSignSGD achieves substantial and consistent accuracy gains over all baselines (Figure 2). These improvements highlight the method's ability to produce models that handle complex, specialized reasoning more effectively, benefiting from faster and more stable optimization.

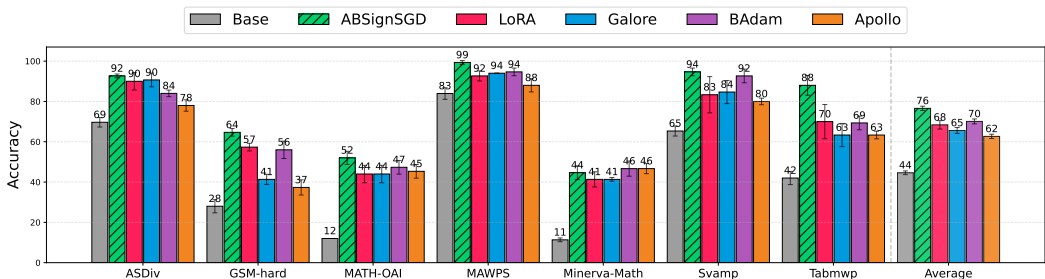

Figure 2: Accuracy on diverse mathematical reasoning benchmarks for Qwen3-8B fine-tuned on OpenMathInstruct-2 with different optimizers. ABSignSGD consistently outperforms baselines across tasks and achieves a $6\%$ accuracy improvement against the second best, indicating that faster convergence during training translates into stronger task-level generalization.

**General Instruction-Following.** On MT-Bench (Table 3), ABSignSGD attains the highest overall average score (6.18) across eight categories, leading in five and remaining highly competitive in the rest. This breadth of strength shows that the method enhances not only domain-specific reasoning but also broad, multi-skill instruction-following capabilities.

Across specialized and general tasks, ABSignSGD consistently surpasses strong memory-efficient baselines. By combining sign-based updates with depth-biased block selection, it accelerates convergence without loss of accuracy, producing high-performing models with limited resources.

Table 3: MT-Bench scores (higher is better) for Qwen3-8B fine-tuned on Stanford-Alpaca with different optimizers. ABSignSGD attains the highest overall average and leads in five categories.

| Method ↑ | Writing | Roleplay | Reasoning | Math | Coding | Extraction | STEM | Humanities | Ave. |
|---|---|---|---|---|---|---|---|---|---|
| Base | 4.64 | 5.04 | 5.12 | 4.93 | 5.07 | 4.66 | 4.88 | 4.59 | 4.87 |
| **Ours** | 5.77 | **6.31** | **6.39** | **6.56** | 6.06 | **6.12** | 6.04 | **6.20** | **6.18** |
| LoRA | **5.82** | 5.81 | 5.34 | 5.87 | 5.37 | 5.77 | 5.48 | 5.66 | 5.64 |
| GaLore | 5.80 | 5.11 | 5.32 | 5.02 | 5.58 | 4.74 | **6.10** | 5.15 | 5.48 |
| BAdam | 5.79 | 5.75 | 6.08 | 5.21 | **6.11** | 5.54 | 5.25 | 6.00 | 5.72 |
| Apollo | 5.63 | 5.43 | 5.51 | 5.56 | 5.11 | 5.45 | 5.08 | 5.43 | 5.40 |

## 4.5 ABLATION STUDY

The preceding results establish ABSignSGD as a powerful, efficient method for fine-tuning LLMs. To identify the key design components behind its success, we perform a thorough ablation study. Unless stated otherwise, all ablations use the smaller **Qwen3-1.7B** model with the same training configuration as the main experiments, reducing computational cost while preserving the relative behavior of variants. Additional results are provided in Appendix F.3.

**Justification for SignSGD.** To isolate the effect of the core optimizer, we compare ABSignSGD with BAdam and BlockSGD (the block-coordinate extension of SGD that updates a single parameter block per iteration) under an identical block-selection scheme with their optimal learning rates, and also include their base optimizers (SignSGD, Adam, and SGD) for reference.

As shown in Figure 3 (Left & Middle), combining a block update strategy with Adam impairs convergence, likely because its adaptive step size relies on past gradient history that block switching erases. In contrast, SGD and SignSGD are not affected as much because they only depend on current gradient information, making them inherently more compatible with the block-coordinate approach.

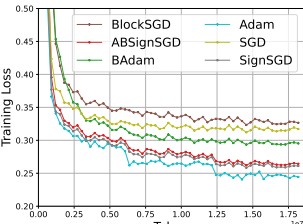 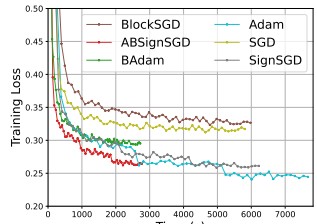 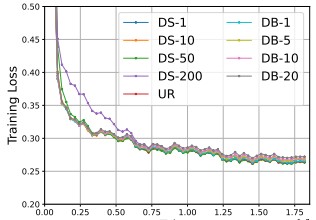

Figure 3: Convergence curves from two ablation experiments on fine-tuning Qwen3-1.7B with OpenMathInstruct-2. **Left & Middle:** Token- and time-wise training loss for core optimizers (SGD, SignSGD, Adam) and their block-update counterparts under an identical block-selection scheme. Adam degrades under block switching, whereas memoryless methods (SGD and SignSGD) remain compatible; SignSGD further outperforms SGD, yielding faster iteration-wise convergence. **Right:** ABSignSGD under different update rules exhibits similar convergence.

Figure 4-Left illustrates why SignSGD converges in our setting: the *sign-agreement probability* distribution is sharply skewed toward 1. Only around 1% of coordinates have agreement probabilities below 0.5, while for a large fraction the sign is almost always correct. This closely matches the method's core assumption and supports its stable convergence behavior.

We now turn to explaining why sign-based methods outperform SGD (Figure 3-Left). Two factors are central. **i) regularization**: prior work shows SignSGD induces an Adam-like regularization effect that is beneficial under heavy class imbalance, which we quantify via the *token-class frequency distribution*. For example, in the Stanford-Alpaca dataset the most frequent token appears roughly $10\times$ more than the second most frequent (Figure 4-Middle). **ii) noise resilience**: the *relative gradient noise magnitude* remains consistently $> 1$ and can exceed $10^3$ (Figure 4-Right); such noise can impede SGD but is naturally damped by sign-based updates.

**On block update rule.** We validate the necessity of our flexible block selection scheme by comparing (1) *depth-biased updates (DB-c)*, where $c$ is the relative backprop. time ratio; (2) *deep-to-shallow selection (DS-K)*, a deterministic scheme updating from the deepest to the shallowest layer

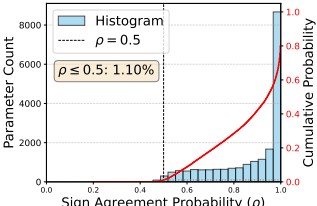 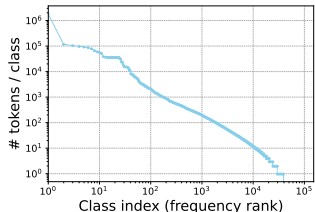 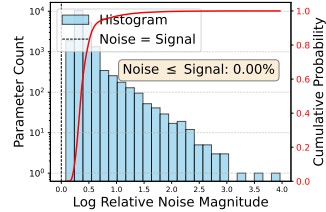

Figure 4: Factors explaining ABSignSGD's success (Qwen3-1.7B on Stanford-Alpaca; details in Appendix F). **Left & Right:** Metrics derived from comparing full gradients with multiple mini-batch gradients over the same parameters. The left panel shows the *sign-agreement probability* histogram, while the right shows the *relative gradient noise magnitude* histogram. **Middle:** *Token-class frequency* distribution, revealing severe class imbalance in the training set.

with block-switching interval $K$; and (3) *uniform random selection (UR)*. Consistent with (Luo et al., 2024), Figure 3-Right shows that, within a moderate hyperparameter range, the selection choice minimally affects convergence. Appendix G.3 further verifies that the choice of scheme has negligible impact on downstream generalization. Consequently, the depth-biased method's primary advantage is efficiency—by updating shallower layers more often, it achieves substantial runtime savings without performance loss.

## 5 CONCLUSION

We introduced ABSignSGD, an arbitrary-order block-coordinate extension of SignSGD for efficient full-parameter fine-tuning of large language models. The framework supports diverse block selection strategies, with the depth-biased scheme as one effective example, and includes a majority-vote (MV) variant for data-parallel training. We provide unified theoretical convergence guarantees for both methods under the *SPB* assumption. Empirical results show that ABSignSGD surpasses strong baselines in convergence speed and downstream accuracy while reducing memory footprint and wall-clock runtime.

## REPRODUCIBILITY STATEMENT

We have taken steps to ensure the reproducibility of our results. All implementation details, including hyperparameters, learning rate schedules, optimizer configurations, dataset splits, evaluation protocols, and hardware specifications, are provided in Appendix D. We also rely on official implementations for baseline methods and evaluation methods to ensure fairness and comparability. Together, these details should enable independent researchers to replicate our experimental findings without ambiguity.

## ETHICS STATEMENT

This work focuses on developing memory- and runtime-efficient optimization algorithms for fine-tuning large language models. All experiments were conducted on publicly available datasets (e.g., OpenMathInstruct-2, Stanford-Alpaca) with open-source models (e.g. Qwen3, Llama3) that do not contain personally identifiable information. We do not foresee direct risks of harm arising from our methodology. Nevertheless, as with any system that improves the efficiency of LLM fine-tuning, there exists the potential for downstream misuse, including generating harmful or biased content. We emphasize that our contributions are intended to advance research in optimization and efficiency, and we encourage responsible and ethical use of the resulting models and techniques.

ACKNOWLEDGMENT

This work was supported in part by the National Natural Science Foundation of China under Grant 62373316 and Grant 62336005, and in part by the 1+1+1 CUHK-CUHK(SZ)-GDSTC Joint Collaboration Fund under Grant 2025A0505000049.

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

# A   MEMORY AND COMMUNICATION ANALYSIS

We first analyze the memory efficiency of different optimizers from two perspectives: block-wise storage during the update of a single block and the global memory footprint across the entire model.

## A.1   BLOCK-WISE STORAGE

Table 4: Block-wise storage requirements (in parameter counts) for different methods during the update of a single block. The table reports the number of parameters that must be stored for weights, gradients, optimizer states, and projection matrices. Precision (e.g., FP16 vs. FP32) is not considered here; actual memory usage can be derived by multiplying these counts by the storage size per parameter.

| Method | Weight | Gradient | Opt. State | Proj. Matrix |
|---|---|---|---|---|
| ABSignSGD | $mn$ | $mn$ | – | – |
| BAdam | $mn$ | $mn$ | $2mn$ | – |
| LoRA | $mn + mr + nr$ | $mr + nr$ | $2mr + 2nr$ | – |
| GaLore | $mn$ | $mn + nr$ | $2nr$ | $mr$ |
| Apollo | $mn$ | $mn + nr$ | $2nr$ | – |

Table 4 reports the storage requirements incurred when updating one block of parameters. The values are expressed in terms of the number of parameters that must be stored (weights, gradients, optimizer states, and projection matrices). These counts are independent of numerical precision; actual memory usage in bytes can be obtained by multiplying the counts by the storage size per parameter (e.g., 2 bytes for FP16, 4 bytes for FP32).

We assume a block weight $W \in \mathbb{R}^{mn}$, with $r$ denoting the low-rank dimension and $m \approx n \gg r$. For all low-rank methods, we assume they use Adam as the base optimizer.

- **ABSignSGD** requires only the block weights and their gradients, with no optimizer states, since updates rely solely on gradient signs.

- **BAdam** stores both gradients and optimizer states just as Adam.

- **LoRA** reduces gradient storage by restricting updates to low-rank matrices. Therefore, it only store two low-rank gradient matrices and their corresponding optimizer states.

- **GaLore** and **Apollo** requires the gradient to conduct low-rank projection. Moreover, GaLore needs to store a projection matrix, while Apollo uses random matrices for projection.

At first glance, low-rank methods appear advantageous because they store only compact matrices. However, block-based methods like ABSignSGD scale more favorably, since they only maintain variables for the currently updated block, whereas low-rank methods must maintain auxiliary states across all blocks.

## A.2   GLOBAL MEMORY FOOTPRINT

Table 5 extends the analysis to the full model (excluding activations), assuming $M$-B parameters and rectangular weights of dimension $m$. We assume the low-rank projection methods to use layer-wise update, as they otherwise require much more memory because they need to store full gradients. $N$ represents the number of blocks or layers for different optimizers. Here we incorporate precision assumptions: we assume model weights to be stored in half-precision, while gradients, optimizer states, and other parameters are maintained in full precision. The total memory footprints are:

- **ABSignSGD:** weights in FP16 plus cached block sign gradients:

$$2M \; + \; \frac{M}{8N}.$$

- **BAdam:** weights in FP16 plus block gradients and optimizer states in FP32 (and an FP32 copy of the block weight for the update):

$$2M + \frac{16M}{N}.$$

- **LoRA:** weights and low-rank matrices in FP16 plus, high-precision copies of the low-ranl matrices, gradients, and optimizer states in FP32:

$$2M + \frac{36Mr}{m}.$$

- **GaLore:** weights in FP16, block gradients and projected gradients in FP32, plus projection/auxiliary terms:

$$2M + \frac{8M}{N} + \frac{12Mr}{m} + \frac{4Mr}{mN}.$$

- **Apollo:** weights in FP16, block gradients and projected gradients in FP32:

$$2M + \frac{8M}{N} + \frac{8Mr}{m} + \frac{4Mr}{mN}.$$

Overall, ABSIGNSGD achieves the lightest footprint by eliminating FP32 optimizer states and using sign-only cached information, while block-wise storage ensures that only the currently updated block contributes transient overhead.

Table 5: Global memory footprint comparison of different methods.

| Method | Weight | Gradient | Opt. State | Other | Total |
|---|---|---|---|---|---|
| ABSignSGD | $2M$ | $\frac{M}{8N}$ | – | – | $2M + \frac{M}{8N}$ |
| BAdam | $2M + \frac{4M}{N}$ | $\frac{4M}{N}$ | $\frac{8M}{N}$ | – | $2M + \frac{16M}{N}$ |
| LoRA | $2M + \frac{12Mr}{m}$ | $\frac{8Mr}{m}$ | $\frac{16Mr}{m}$ | – | $2M + \frac{36Mr}{m}$ |
| GaLore | $2M + \frac{4M}{N}$ | $\frac{4M}{N} + \frac{4Mr}{mN}$ | $\frac{8Mr}{m}$ | $\frac{4Mr}{m}$ | $2M + \frac{8M}{N} + \frac{12Mr}{m} + \frac{4Mr}{mN}$ |
| Apollo | $2M + \frac{4M}{N}$ | $\frac{4M}{N} + \frac{4Mr}{mN}$ | $\frac{8Mr}{m}$ | – | $2M + \frac{8M}{N} + \frac{8Mr}{m} + \frac{4Mr}{mN}$ |

### A.3 COMMUNICATION BUDGET

In distributed implementations, the communication budget is determined by the amount of gradient information that must be exchanged across workers at each synchronization step.

For ABSIGNSGD-MV, BADAM, and LORA, the communication cost is directly proportional to their gradient storage requirements in Table 5. That is, each worker must transmit the same number of parameters as it stores locally for gradients, ensuring consistency across replicas.

In contrast, GALORE and APOLLO require transmitting the *full gradient* of size $4M$ to maintain mathematical equivalence during distributed updates. This communication volume is orders of magnitude larger than that of the other methods, and can quickly become the dominant bottleneck in multi-GPU or multi-node training.

While it might be possible to reduce this overhead by transmitting only the low-rank projected gradient of size $\frac{4Mr}{mN}$, such a modification would alter the update rule and falls outside the scope of this paper.

## B CONVERGENCE PROOF

### B.1 AUXILIARY LEMMAS

**Lemma B.1.** *The alignment norm $\|\cdot\|_{\mathcal{N}}$ satisfies the following conditions*

- *Triangular inequality:* $\|x - y\|_{\mathcal{N}} \geq \|x\|_{\mathcal{N}} - \|y\|_{\mathcal{N}}, \forall x, y.$

- *Upper bounded by $l_1$ norm:* $\|x\|_{\mathcal{N}} \leq \|x\|_1, \forall x.$

*Proof.* Note that

$$
\begin{aligned}
\|x\|_{\mathcal{N}} - \|y\|_{\mathcal{N}} &= w_i |x_i| - w_i |y_i| \\
&\leq w_i |x_i - y_i| \\
&= \|x - y\|_{\mathcal{N}},
\end{aligned}
$$

and

$$
\begin{aligned}
\|x\|_{\mathcal{N}} &= \sum_i w_i |x_i| \\
&\leq \sum_i |x_i| \\
&= \|x\|_1.
\end{aligned}
$$

**Lemma B.2.** *For all $x \in \mathbb{R}^d$, $l_1$-norm and $l_2$-norm have the following relation*

$$
\|x\|_1 \leq \sqrt{d}\|x\|_2.
$$

$\square$

## B.2 PROOF OF THEOREM 3.4

The proof is an extension of the one in (Safaryan & Richtárik, 2021) to the block-update scheme. We first provide a block descent lemma, which arises naturally after the definition of the customized norm $\|\cdot\|_{\rho}$.

**Lemma B.3.** *[Block descent lemma] Given Assumption 3.1 and 3.2 , and assuming identical block sizes, the updates of ABSignSGD and ABSignSGD-MV satisfy*

$$
\frac{\sum_{k=0}^{K-1} \mathbb{E}\|\nabla_{\mathbf{i_k}} f(x^k)\|_{\mathcal{N}}}{K} \leq \frac{f(x^0) - f^*}{\alpha K} + \frac{\alpha dL}{2N}. \tag{4}
$$

*Proof.* We first prove for the case of ABSignSGD. By $L$-smoothness of the objective function $f(\cdot)$,

$$
\begin{aligned}
f(x^{k+1}) - f(x^k) &\leq \nabla f(x^k)^T \cdot (x^{k+1} - x^k) + \frac{L}{2}\|x^{k+1} - x^k\|_2^2 \\
&= -\alpha \nabla_{\mathbf{i_k}} f(x^k)^T \cdot \text{sign}(g_{\mathbf{i_k}}(x^k)) + \frac{\alpha^2 d_{i_k} L}{2}.
\end{aligned}
$$

Note that by the definition of sign-alignment probabilities $\rho_i$, we get

$$
\begin{aligned}
\mathbb{E}[\nabla_{\mathbf{i_k}} f(x^k)^T \cdot \text{sign}(g_{\mathbf{i_k}}(x^k))] &= \nabla_{\mathbf{i_k}} f(x^k)^T \cdot \mathbb{E}[\text{sign}(g_{\mathbf{i_k}}(x^k))] \\
&= \sum_{i \in \pi_{i_k}} \nabla_i f(x^k)(\rho_i(x^k)\text{sign}(\nabla_i f(x^k)) - (1 - \rho_i(x^k))\text{sign}(\nabla_i f(x^k)) \\
&= \sum_{i \in \pi_{i_k}} (2\rho_i(x_k) - 1)|\nabla_i f(x^k)| \\
&= \|\nabla_{\mathbf{i_k}} f(x^k)\|_{\mathcal{N}}.
\end{aligned}
$$

Combining the above and take full expectation, we arrive at

$$
\mathbb{E}\|\nabla_{\mathbf{i_k}} f(x^k)\|_{\mathcal{N}} \leq \frac{f(x^k) - f(x^{k+1})}{\alpha} + \frac{\alpha d_{i_k} L}{2}. \tag{5}
$$

For ABSignSGD-MV, we define the majority vote of block gradient signs as

$$
\hat{g}_{\mathbf{i_k}}(x^k) := \sum_{j=1}^n \text{sign}(g_{\mathbf{i_k}}^j(x^k)).
$$

Update (3) can be rewritten as

$$x_{\mathbf{i_k}}^{k+1} = x_{\mathbf{i_k}}^k - \alpha \cdot \text{sign}(\hat{g}_{\mathbf{i_k}}(x^k)).$$

Following an identical reasoning as in ABSignSGD,

$$f(x^{k+1}) - f(x^k) \leq -\alpha \nabla_{\mathbf{i_k}} f(x^k)^T \cdot \text{sign}(\hat{g}_{\mathbf{i_k}}(x^k)) + \frac{\alpha^2 d_{i_k} L}{2}. \tag{6}$$

By Lemma 13 in (Safaryan & Richtárik, 2021), we have

$$
\begin{aligned}
\mathbb{E}[\nabla_{\mathbf{i_k}} f(x^k)^T \cdot \text{sign}(\hat{g}_{\mathbf{i_k}}(x^k))] &= \nabla_{\mathbf{i_k}} f(x^k)^T \cdot \mathbb{E}[\text{sign}(\hat{g}_{\mathbf{i_k}}(x^k))] \\
&= \sum_{\substack{1 \leq i \leq d \\ \nabla_{\mathbf{i_k}} f(x^k) \neq 0}} \left| \nabla_{\mathbf{i_k}} f(x^k) \right| \cdot \mathbb{E}\left[\text{sign}\left(\hat{g}_{\mathbf{i_k}}(x^k) \cdot \nabla_{\mathbf{i_k}} f(x^k)\right)\right] \\
&= \sum_{\substack{1 \leq i \leq d \\ \nabla_{\mathbf{i_k}} f(x^k) \neq 0}} \left| \nabla_{\mathbf{i_k}} f(x^k) \right| (2I(\rho_i(x_k);\, l,\, l) - 1) \\
&= \|\nabla_{\mathbf{i_k}} f(x^k)\|_{\mathcal{N}}.
\end{aligned}
$$

Combining equation (6) with the above, we reached equation (5).

Sum over equation (5) over $K$ iterations, we have

$$
\begin{aligned}
\frac{\sum_{k=0}^{K-1} \mathbb{E}\|\nabla_{\mathbf{i_k}} f(x^k)\|_{\mathcal{N}}}{K} &\leq \frac{f(x^0) - f^*}{\alpha K} + \frac{\alpha L \sum_{k=0}^{K-1} d_{i_k}}{2K} \\
&= \frac{f(x^0) - f^*}{\alpha K} + \frac{\alpha d L}{2N}.
\end{aligned}
$$

$\square$

Lemma B.3 states that the average of block gradient alignment norm converges under a proper step size. Now, we analyze the relation between the block gradient and the full gradient to show the convergence of the latter.

**Lemma B.4.** *Given identical assumptions in Lemma B.3 and Assumption 3.3, the updates of AB-SignSGD and ABSignSGD-MV satisfy*

$$\|\nabla f(x^k)\|_{\mathcal{N}} - \sum_{t \in S_k} \|\nabla_{\mathbf{i_t}} f(x^t)\|_{\mathcal{N}} \leq L\alpha d (B - \frac{N+1}{2}) \tag{7}$$

*Proof.* From Assumption 3.3, each block is updated at least once for every $B$ iterations. Therefore, we pick $N$ steps in which each step updates a different block and we have

$$\sum_{t=k}^{k+B-1} \|\nabla_{\mathbf{i_t}} f(x^t)\|_{\mathcal{N}} \geq \sum_{t \in S_t} \|\nabla_{\mathbf{i_t}} f(x^t)\|_{\mathcal{N}},$$

where $S_t \subseteq \{k, ..., k+B-1\}$, $|S_t| = N$, and $\{i_t\}_{t \in S_t} = [N]$.

We compare the RHS to the full-gradient at step $k$ as follows

$$
\begin{aligned}
\|\nabla f(x^k)\|_{\mathcal{N}} - \sum_{t \in S_k} \|\nabla_{\mathbf{i_t}} f(x^t)\|_{\mathcal{N}} &= \sum_{i=1}^{N} \left( \|\nabla_{\mathbf{i}} f(x^k)\|_{\mathcal{N}} - \|\nabla_{\mathbf{i}} f(x^{k+t_i-1})\|_{\mathcal{N}} \right) \\
&\leq \sum_{i=1}^{N} \left( \|\nabla_{\mathbf{i}} f(x^k) - \nabla_{\mathbf{i}} f(x^{k+t_i-1})\|_{\mathcal{N}} \right) \\
&\leq \sum_{i=1}^{N} \left( \|\nabla_{\mathbf{i}} f(x^k) - \nabla_{\mathbf{i}} f(x^{k+t_i-1})\|_{1} \right) \\
&\leq \sum_{i=1}^{N} \left( \sqrt{d_i} \|\nabla_{\mathbf{i}} f(x^k) - \nabla_{\mathbf{i}} f(x^{k+t_i-1})\|_{2} \right) \\
&\leq \sum_{i=1}^{N} \left( \sqrt{d_i} \|\nabla f(x^k) - \nabla f(x^{k+t_i-1})\|_{2} \right) \\
&\leq \sum_{i=1}^{N} \left( \sqrt{d_i} L \|x^k - x^{k+t_i-1}\|_{2} \right)
\end{aligned}
$$

where the updating block at $k + t_i - 1$ is the $i$-th block.

Note that the second last inequality above is necessary, as the difference between $x^k$ and $x^{k+t_i-1}$ crosses multiple blocks, so

$$
\|\nabla_{\mathbf{i}} f(x^k) - \nabla_{\mathbf{i}} f(x^{k+t_i-1})\|_2 \leq L \|x_{\mathbf{i}}^k - x_{\mathbf{i}}^{k+t_i-1}\|_2
$$

does not hold.

Note that

$$
\begin{aligned}
\sum_{i=1}^{N} \left( \sqrt{d_i} L \|x^k - x^{k+t_i-1}\|_2 \right) &\leq \sum_{i=1}^{N} L \sqrt{\frac{d}{N}} \left( \sum_{j=k}^{k+t_i-2} \|x^{j+1} - x^j\|_2 \right) \\
&\leq \sum_{i=1}^{N} L \sqrt{\frac{d}{N}} \left( \sum_{j=k}^{k+t_i-2} \alpha \sqrt{\frac{d}{N}} \right) \\
&\leq L \alpha \frac{d}{N} \sum_{i=1}^{N} (t_i - 1).
\end{aligned}
$$

Block $i$ updates at iteration $k + t_i - 1$. However, there is only one block updating at each iteration. The worst case is when all blocks update once within the last $N$ iterations in the $[k, k+B-1]$ window. Therefore,

$$
\begin{aligned}
\sum_{i=1}^{N} (t_i - 1) &\leq \sum_{j=0}^{N-1} (B - N + j) \\
&= N \left( B - \frac{N+1}{2} \right)
\end{aligned}
$$

Now we bound the staleness error, or the distance traveled between the $t_i$ steps

$$
\begin{aligned}
\|x^k - x^{k+t_i-1}\|_2 &\leq \sum_{t=k}^{k+t_i} \|x^{t+1} - x^t\|_2 \\
&= \sum_{t=k}^{k+t_i} \alpha \sqrt{d_{i_t}} \\
&\leq \alpha B \sqrt{\frac{d}{N}}.
\end{aligned}
$$

Combining the above, we arrived at the conclusion. □

Theorem 3.4 is a straightforward combination of Lemma B.3 and B.4.

Sum equation (7) over $KB$ iterations, we have

$$\sum_{k=0}^{K-1} \|\nabla f(x^{kB})\|_{\mathcal{N}} \leq \sum_{t=0}^{(K-1)B} \|\nabla_{\mathbf{i_t}} f(x^t)\|_{\mathcal{N}} + \alpha L d K (B - \frac{N+1}{2}).$$

Combine the above with equation (4), we have

$$\frac{\sum_{k=0}^{K-1} \mathbb{E}\|\nabla f(x^{kB})\|_{\mathcal{N}}}{K} \leq \frac{\sum_{t=0}^{(K-1)B} \mathbb{E}\|\nabla_{\mathbf{i_t}} f(x^t)\|_{\mathcal{N}}}{K} + \alpha L B d$$

$$\leq \frac{f(x^0) - f^*}{\alpha K} + \alpha L d \big(B\big(1 + \frac{1}{2N}\big) - \frac{N+1}{2}\big)\big).$$

## C ROBUSTNESS OF MAJORITY VOTING

Safaryan & Richtárik (2021) states that the Majority Vote (MV) estimator's error rate converges exponentially to 0. But we are not sure how the MV estimate compared to averaging the gradient. Here we prove that under heavy-tailed noise, the MV estimator is asymptotically infinitely more accurate than the standard Summation (or Averaging) estimator.

### C.1 PROBLEM SETTING

With a slight abuse of notation, we consider the problem of estimating the sign of a true signal $f \in \mathbb{R}$ from a set of $M$ independent observations. Each observation is generated from a signal-plus-noise model:

$$g_i = f + \epsilon_i, \quad \text{for } i = 1, \ldots, M,$$

where $\{\epsilon_i\}_{i=1}^M$ are i.i.d. random noise variables with $\mathbb{E}[\epsilon_i] = 0$. We assume each observation provides a weak but better-than-random signal about the sign of $f$. This is captured by the per-worker accuracy $\rho$:

$$\rho := \mathbb{P}(\text{sign}(g_i) = \text{sign}(f)) > 1/2.$$

We analyze and compare two sign estimators:

1. **Majority Vote (MV):** The sign is estimated by aggregating the signs of the individual observations. Let $S_i = \text{sign}(g_i) \in \{-1, +1\}$.

$$\hat{f}_{\text{MV}} = \text{sign}\left(\sum_{i=1}^M S_i\right).$$

2. **Summation (SUM):** The sign is estimated from the sum of the raw observations.

$$\hat{f}_{\text{SUM}} = \text{sign}\left(\sum_{i=1}^M g_i\right).$$

Without loss of generality, let $f > 0$. The error probabilities are then $\text{PMV}(M) = \mathbb{P}(\sum S_i \leq 0)$ and $\text{PSUM}(M) = \mathbb{P}(\sum g_i \leq 0) = \mathbb{P}(\sum \epsilon_i \leq -Mf)$.

### C.2 ASYMPTOTIC ANALYSIS OF ESTIMATOR ERROR

We first establish the benchmark performance of the MV estimator, which is known to be robust regardless of the noise distribution's tail behavior.

**Theorem C.1** (MV Error Bound). *The error probability of the Majority Vote estimator converges exponentially to zero:*

$$PMV(M) \leq \exp(-CM),$$

*where $C = (2\rho - 1)^2/2$ is a positive constant.*

*Proof.* The result is a direct application of Hoeffding's inequality (Hoeffding, 1963) to the sum of i.i.d. bounded variables $S_i \in [-1, 1]$ with positive mean $\mathbb{E}[S_i] = 2\rho - 1 > 0$. $\qquad \square$

Next, we analyze the SUM estimator under the condition of heavy-tailed noise, modeled as a regularly varying distribution.

**Assumption C.2** (Heavy-Tailed Noise). *The noise distribution has a regularly varying left tail. Let* $Y_i = -\epsilon_i$. *The tail probability* $\mathbb{P}(Y_i > t)$ *is regularly varying at infinity with index* $\alpha > 1$:

$$\mathbb{P}(Y_i > t) = t^{-\alpha}L(t),$$

*where* $L(t)$ *is a slowly varying function.*

Under this assumption, the SUM estimator's performance degrades significantly. The following theorem formalizes this by showing that its error, relative to the MV estimator, diverges.

**Theorem C.3** (Asymptotic Dominance of Majority Vote). *Under Assumption C.2, the ratio of the error probabilities for the Summation and Majority Vote estimators diverges to infinity:*

$$\frac{PSUM(M)}{PMV(M)} \xrightarrow{M \to \infty} \infty.$$

*Proof.* We establish an asymptotic lower bound for the ratio. The error of the SUM estimator is determined by the "big-jump principle" for subexponential distributions (Embrechts et al., 2013), which states $\text{PSUM}(M) \sim M \cdot \mathbb{P}(Y_1 > Mf)$ when $Mf \to \infty$. Under Assumption C.2, this yields the asymptotic lower bound:

$$\text{PSUM}(M) \gtrsim c_1 M^{1-\alpha} f^{-\alpha} L(Mf),$$

for some constant $c_1 > 0$. Combining this with the upper bound for the MV error, we have for all sufficiently large $M$:

$$\frac{\text{PSUM}(M)}{\text{PMV}(M)} \gtrsim \frac{c_1 M^{1-\alpha} f^{-\alpha} L(Mf)}{\exp(-CM)}$$
$$= c_1 f^{-\alpha} \cdot \exp(CM) \cdot M^{-(\alpha-1)} \cdot L(Mf).$$

The limit of this expression as $M \to \infty$ is determined by the competition between the exponential growth term, $\exp(CM)$, and the product of the polynomial decay term, $M^{-(\alpha-1)}$, and the slowly varying term, $L(Mf)$. As exponential growth dominates both polynomial decay and slowly varying functions, the lower bound diverges to infinity. Consequently, the ratio itself must also diverge to infinity. This demonstrates that the MV estimator is asymptotically infinitely more accurate than the SUM estimator in this setting. $\qquad \square$

## D   DETAILS ON EXPERIMENTAL SETUP

To ensure a fair and reproducible comparison across all methods, we standardized the training and evaluation pipeline. Below we describe the configuration in detail.

### D.1   LEARNING RATE SCHEDULE AND HYPERPARAMETER SEARCH

All methods adopt a **linear learning rate schedule** with a warmup phase covering the first 10% of total training steps. Learning rates were selected via a **logarithmic grid search** in the range $3 \times 10^{-7}$ to $1 \times 10^{-3}$, with a multiplicative step size of 3. The chosen learning rates for each method–model pair are summarized in Table 6.

### D.2   DETAILS ON DEPTH-BIASED UPDATE RULE

**Derivation of the Bounded Update Interval** $B$**.**   Recall that our depth-biased rule selects the block $i$ with the minimum timestamp $T_i$ and updates it as $T_i \leftarrow T_i + \tau_i$. This is equivalent to a weighted round-robin schedule where the frequency of block $i$ is proportional to $1/\tau_i$.

| Model | ABSignSGD | LoRA | GaLore | BAdam | Apollo |
|---|---|---|---|---|---|
| Qwen3-8B-Math | 3e-5 | 3e-6 | 1e-5 | 1e-5 | 1e-5 |
| Qwen3-8B-Alpaca | 3e-5 | 1e-5 | 1e-4 | 1e-5 | 1e-5 |
| Llama3-8B-Math | 1e-5 | 3e-6 | 3e-5 | 3e-6 | 3e-5 |
| Llama3-8B-Alpaca | 1e-5 | 3e-6 | 3e-5 | 3e-6 | 3e-5 |
| Llama3-32B-Math | 3e-5 | 1e-6 | 3e-5 | 1e-5 | 1e-5 |

Table 6: Final learning rates for each method across models.

To satisfy Assumption 3.3, we must show there exists a finite integer $B$ such that every block is visited at least once in any window of $B$ iterations. Consider the "slowest" block $i_{slow}$ which has the largest virtual cost $\tau_{max} = \max_i \tau_i$. In the worst-case scenario, between two consecutive updates of block $i_{slow}$, any other block $j$ can be updated at most $\lceil \tau_{max}/\tau_j \rceil$ times. Therefore, the maximum number of iterations between two updates of any block is bounded by the sum of these worst-case update counts:

$$B = \sum_{j=1}^{N} \left\lceil \frac{\tau_{max}}{\tau_j} \right\rceil \tag{8}$$

Given that all $\tau_j \geq 1$ and $N$ is finite, $B$ is finite, thereby satisfying the bounded update interval assumption.

**Execution Trace.** To clarify the mechanism described in Section 4.1, we provide a concrete example of the depth-biased selection strategy. This strategy is purely algorithmic and does not depend on real-time hardware measurements.

Consider a simplified model with $N = 4$ blocks. We define the virtual update cost $\tau_i$ such that deeper blocks have lower costs (higher frequency). Let us assume the costs are calculated as $\boldsymbol{\tau} = [6, 5, 4, 3]$ for blocks 1 through 4 respectively (where Block 4 is the deepest).

The algorithm maintains a virtual timestamp vector $\mathbf{T}$, initialized as $\mathbf{T} = \boldsymbol{\tau}$. At every step, the block with the lowest $T_i$ is selected, and its value is incremented by its cost $\tau_i$.

Table 7: Trace of block selection over the first 5 steps.

| Step ($k$) | State vector T (before selection) | Min Value | Selected Block ($i_k$) | Action $(T_{i_k} \leftarrow T_{i_k} + \tau_{i_k})$ |
|---|---|---|---|---|
| 0 | $[6, 5, 4, \mathbf{3}]$ | 3 | **Block 4** | $T_4 \leftarrow 3 + 3 = 6$ |
| 1 | $[6, 5, \mathbf{4}, 6]$ | 4 | **Block 3** | $T_3 \leftarrow 4 + 4 = 8$ |
| 2 | $[6, \mathbf{5}, 8, 6]$ | 5 | **Block 2** | $T_2 \leftarrow 5 + 5 = 10$ |
| 3 | $[\mathbf{6}, 10, 8, 6]$ | 6 | **Block 1**[*] | $T_1 \leftarrow 6 + 6 = 12$ |
| 4 | $[12, 10, 8, \mathbf{6}]$ | 6 | **Block 4** | $T_4 \leftarrow 6 + 3 = 9$ |

[*]Note: In the event of a tie (as seen in Step 3 where $T_1 = 6$ and $T_4 = 6$), we prioritize the shallower block to ensure coverage, though any consistent tie-breaking rule works.

As shown, Block 4 (the "fastest" or deepest block) is updated again at Step 4, while Block 2 has not yet been revisited. Over many iterations, the update count for block $i$ converges to be proportional to $1/\tau_i$.

### D.3 OPTIMIZERS AND PRECISION SETTINGS

- **AdamW**: Used whenever "Adam" is referenced, with PyTorch default hyperparameters $(\beta_1, \beta_2, \epsilon)$.
- **LoRA**: Configured with rank $r = 8$, consistent with its common lightweight adapter setting.
- **Low-Rank Projection Methods (GaLore, Apollo, Flora)**: Configured with rank $r = 128$, aligning with their reported accuracy–efficiency trade-off.

- **Precision**:
  - ABSignSGD run in **half-precision** as it does not benefit from mixed-precision training due to its sign-based update rule.
  - Low-rank projection methods (GaLore, Apollo) run in **half-precision** following their official implementations.
  - LoRA and BAdam follow their official/popular implementations, using **mixed precision** training.

## D.4 TRAINING CONFIGURATION

- Epochs: All models are trained for 3 epochs, with the final checkpoint used for evaluation.

- Batch size = 16, sequence length = 128.

- Gradient checkpointing: Enabled.

- Gradient accumulation: Disabled. Note that ABSignSGD, BAdam, and LoRA are compatible with gradient accumulation for much larger batches. But this option is off for a fair comparison.

- Layer-wise updates: Low-rank projection methods (GaLore, Apollo) employ **layer-wise updates** for maximum memory save, which are incompatible with standard gradient accumulation.

- Trainable weights: All methods update only the transformer layers leaving other weights intact.

- Offloading: Disabled.

- Quantization: Disabled.

## D.5 HARDWARE ENVIRONMENT

- GPU: All experiments are conducted on a single NVIDIA RTX 3090 GPU (24 GB VRAM).

- CPU: Intel(R) Xeon(R) Silver 4310 @ 2.10GHz.

- Repetition: Each configuration is repeated 3 times with different random seeds; we report mean $\pm$ standard deviation.

- Isolation: No GPU interconnect is used, ensuring results are not influenced by distributed hardware variability.

## D.6 EVALUATION PROTOCOLS

- **MT-Bench**: Evaluated using GPT-5 as a judge via `FastChat`, with fixed decoding parameters (temperature = 0, top-p = 1, fixed seed, official prompt templates).

- **Math-Eval-Harness**: Official dataset splits with standardized prompt formatting and decoding parameters (temperature = 0, top-p = 1, max new tokens = 256, fixed seeds).

## D.7 RUNTIME AND MEMORY MEASUREMENT

- Runtime: Measured using `time.time()` and `torch.cuda.synchronize()` before and after the timed region.

- Memory: Record peak reserved GPU memory using `torch.cuda.max_memory_reserved()` after calling `torch.cuda.reset_peak_memory_stats()`.

## D.8 BASELINE IMPLEMENTATIONS

To ensure reproducibility, we rely on the official implementations of baseline methods without modification:

| Method | Official Repository |
|---|---|
| LoRA (PEFT) | `https://github.com/huggingface/peft` |
| GaLore | `https://github.com/jiaweizhao/GaLore` |
| BAdam | `https://github.com/Ledzy/BAdam` |
| Apollo | `https://github.com/zhuhanqing/APOLLO` |

Table 8: Official implementation links for baseline methods.

# E   ADDITIONAL RESULTS FOR MAIN EXPERIMENTS

## E.1   CONVERGENCE

We provide extended convergence curves for Qwen3-8B on Stanford-Alpaca, Llama3-8B on both OpenMathInstruct-2 and Stanford-Alpaca, and Qwen3-32B on OpenMathInstruct-2 (see Figure 5). Across all tasks, ABSignSGD demonstrates consistently faster iteration-wise convergence compared to LoRA, GaLore, BAdam, and Apollo. Moreover, ABSignSGD's lead widens with respect to time. These findings reinforce the claim that the sign-based block updates are particularly well-suited for large-scale fine-tuning, where both iteration count and runtime are critical bottlenecks.

## E.2   DOWNSTREAM PERFORMANCE

We next evaluate downstream task performance of Llama3-8B and Qwen3-32B fine-tuned with different optimizers on mathematical reasoning (See Figure 6) and general instruction-following (see Table 9) benchmarks. The trends are similar to those of the main text.

Table 9: MT-Bench scores (higher is better) for Llama3-8B fine-tuned on Stanford-Alpaca with different optimizers. ABSignSGD attains the highest overall average and leads in four categories, while remaining competitive in the rest.

| Method ↑ | Writing | Roleplay | Reasoning | Math | Coding | Extraction | STEM | Humanities | Ave. |
|---|---|---|---|---|---|---|---|---|---|
| Base | 4.85 | 5.37 | 4.92 | 4.72 | 4.65 | 4.57 | 4.49 | 4.87 | 4.81 |
| **Ours** | 5.52 | **6.35** | **6.15** | **6.39** | 5.90 | 5.71 | **6.22** | 5.66 | **5.99** |
| LoRA | 5.40 | 6.03 | 5.29 | 5.39 | 5.08 | 4.71 | 5.31 | 5.16 | 5.30 |
| GaLore | 5.17 | 5.76 | 5.06 | 5.35 | 5.50 | 5.40 | 6.01 | **6.30** | 5.57 |
| BAdam | 5.35 | 5.72 | 6.03 | 5.78 | 4.99 | **6.38** | 5.48 | 5.19 | 5.62 |
| Apollo | **5.58** | 5.78 | 5.60 | 6.09 | **5.92** | 5.78 | 5.23 | 5.37 | 5.67 |

# F   DETAILS ON ABLATION STUDY

In this section, we detail how the ablation statistics in Figure 4 were computed and how to interpret them. We also include additional results across more models and datasets, demonstrating the generality of our findings.

## F.1   METHODOLOGY

We create a set $S_p$ of 20,000 parameters uniformly sampled across transformer layers (attention, MLP, embeddings). For each parameter $i \in S_p$, we compute (i) the full-batch gradient $\nabla_i f$ by aggregating over the entire training set and (ii) 500 mini-batch gradients $\{g_i^j\}_{j=1}^{500}$ from randomly sampled mini-batches.

From these, we derive:

1. **Sign agreement probability**

$$\rho_i = \frac{1}{500} \sum_{j=1}^{500} \mathbf{1}\Big\{ \mathrm{sign}(g_i^j) = \mathrm{sign}(\nabla_i f) \Big\},$$

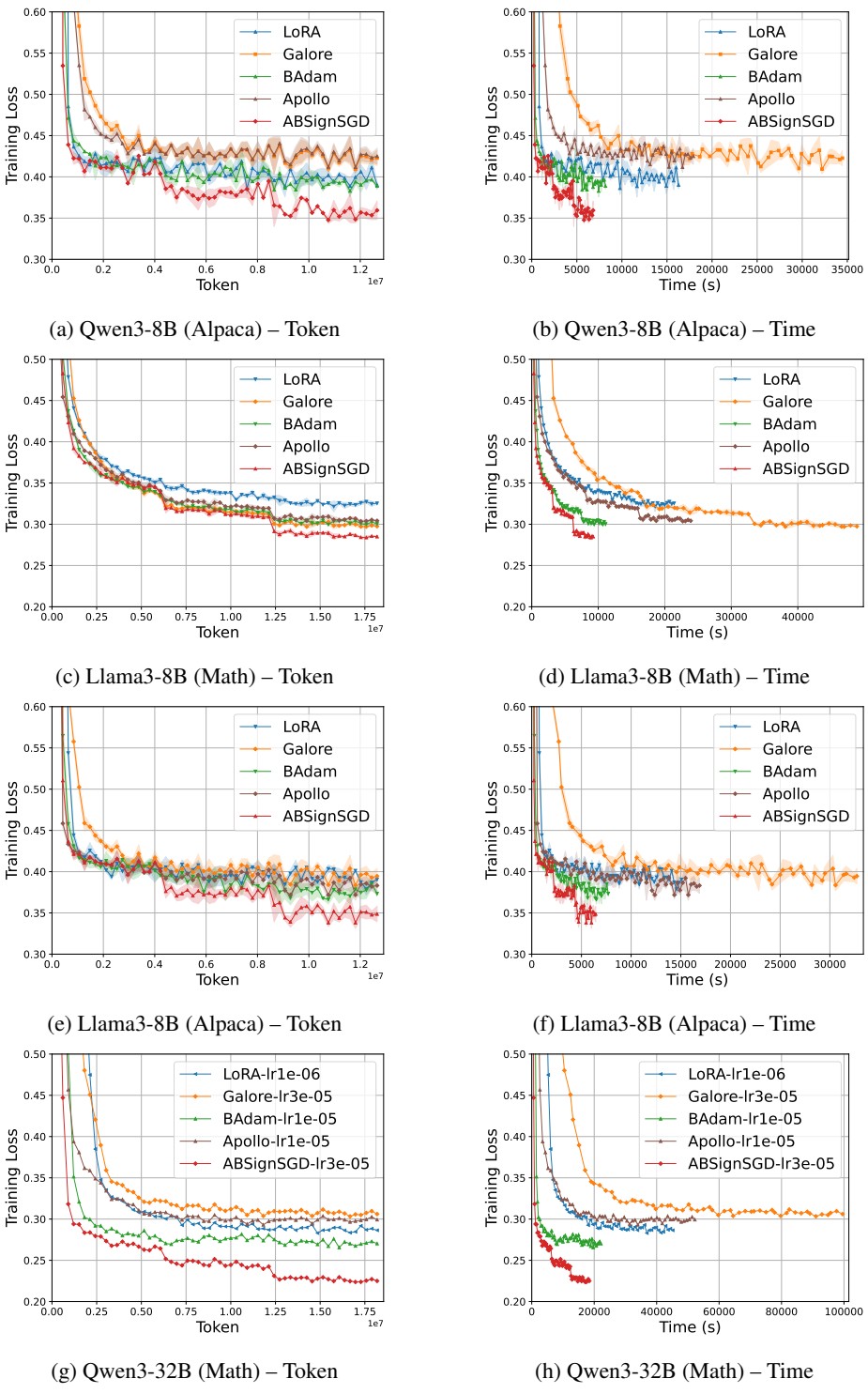

(a) Qwen3-8B (Alpaca) – Token

(b) Qwen3-8B (Alpaca) – Time

(c) Llama3-8B (Math) – Token

(d) Llama3-8B (Math) – Time

(e) Llama3-8B (Alpaca) – Token

(f) Llama3-8B (Alpaca) – Time

(g) Qwen3-32B (Math) – Token

(h) Qwen3-32B (Math) – Time

Figure 5: Additional Training Loss Curves. Left column: Loss vs. Tokens. Right column: Loss vs. Wall-clock Time. Rows represent different Model/Task combinations. ABsignSGD consistently outperforms baselines, with the advantage widening in terms of wall-clock time.

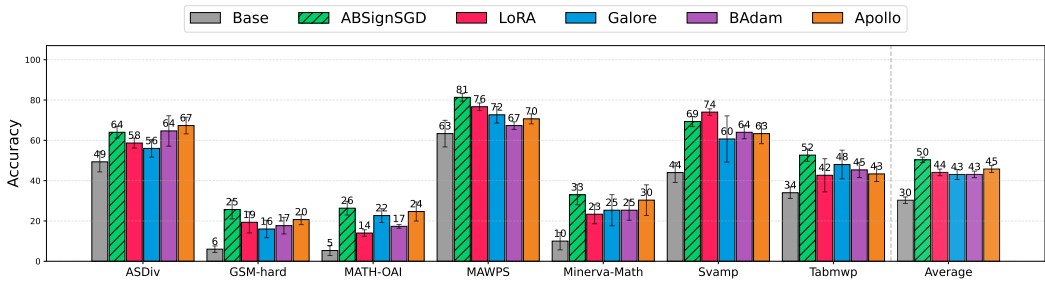

(a) Llama3-8B on OpenMathInstruct-2

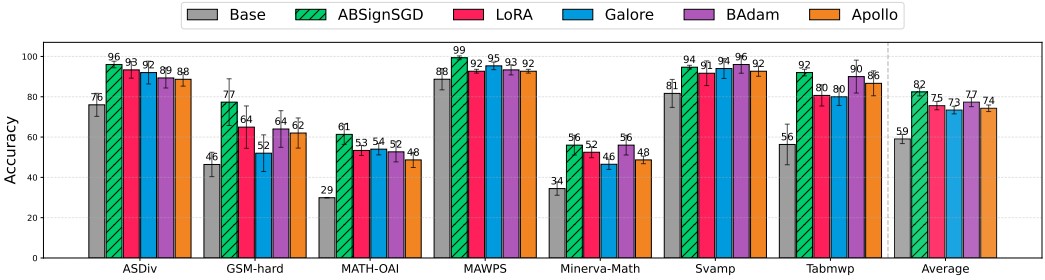

(b) Qwen3-32B on OpenMathInstruct-2

Figure 6: Accuracy on diverse mathematical reasoning benchmarks. **Top:** Llama3-8B results. **Bottom:** Qwen3-32B results. In both scales, ABSignSGD achieves a significant accuracy improvement against the second-best baseline (e.g., $5\%$ improvement for both models).

which measures how reliably the mini-batch gradients align in direction with the full gradient.

2. **Relative noise magnitude**

$$\eta_i = \frac{1}{500\,|\nabla_i f|}\sum_{j=1}^{500}\left|g_i^j - \nabla_i f\right|,$$

which quantifies the average deviation of the mini-batch gradients relative to signal strength.

Figure 4-Left and -Right plot the distributions of $\rho_i$ and $\eta_i$, respectively. Figure 4-Middle shows the class index histogram for all tokens in the training set, which exhibits a pronounced long-tail distribution.

### F.2 KEY OBSERVATIONS

Across all primary settings, we observe:

- **Stable signs:** Most coordinates have large $\rho_i$ (e.g. $> 0.7$), indicating that gradient signs provide meaning information to guide training. This matches Assumption 3.2 for theoretical analysis.

- **Long-tailed token frequencies:** Class frequency histograms show a heavy head and long tail, evidencing severe class imbalance.

- **Noise-dominated magnitudes:** $\eta_i$ is typically $\gg 1$, implying that raw gradient magnitudes are unreliable compared to their signs.

### F.3 ADDITIONAL RESULTS ACROSS MODELS AND DATASETS

To assess robustness and demonstrate generality, we extend the ablation to additional model–dataset pairs, including **Qwen3-1.7B**, **Llama-3.2-1B**, and **GPT-Neo-1.3B** on **OpenMathInstruct-2** and **Stanford-Alpaca**. As shown in Figure 7, the same qualitative patterns persist across architectures and corpora: gradient signs remain highly reliable, token distributions are strongly imbalanced, and noise magnitudes exceed signal strength in most coordinates.

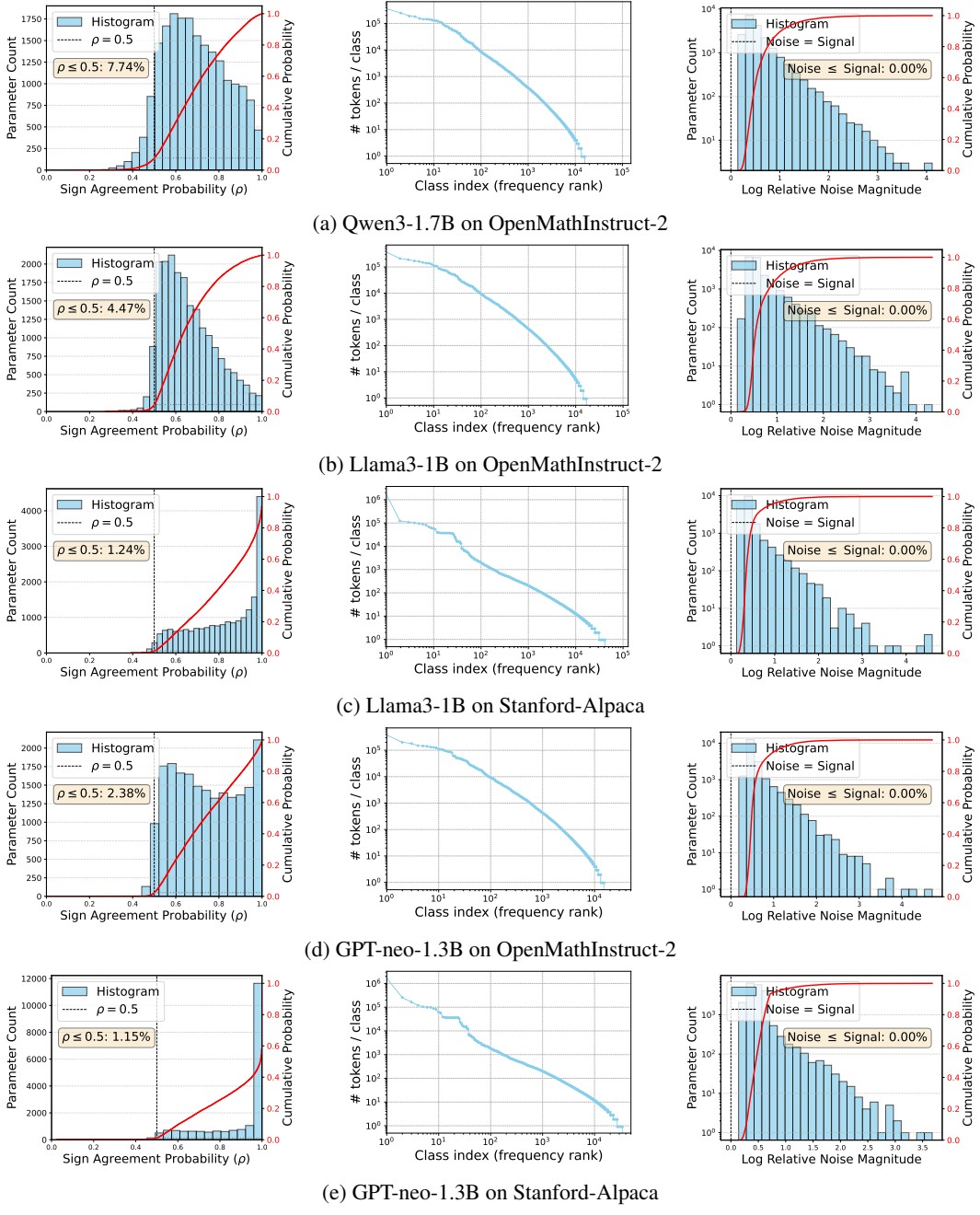

Figure 7: Additional ablation results across model–dataset pairs. The qualitative trends match those in Figure 4.

## G ROBUSTNESS AND SCALABILITY ANALYSIS

In this section, we provide additional empirical results assessing the robustness of ABSignSGD under diverse conditions. Specifically, we evaluate: (1) the method's sensitivity to extreme noise via batch size reduction, (2) its scaling behavior in distributed settings, and (3) the impact of different block-switching schemes on downstream generalization.

### G.1 NOISE SENSITIVITY (BATCH SIZE LIMITS)

As discussed in Section 2.2.3, sign-based methods can theoretically diverge if the sign-agreement probability drops below 0.5. To empirically test this limit, we fine-tune QWEN3-8B on the OpenMathInstruct-2 dataset while decreasing the batch size from 16 down to 4. To strictly isolate the effects of noise, we fix the learning rate to the optimal value identified for the baseline configuration (batch size 16, non-MV) across all experimental runs. As shown in Figure 8, although the convergence speed degrades as noise increases, the method remains stable.

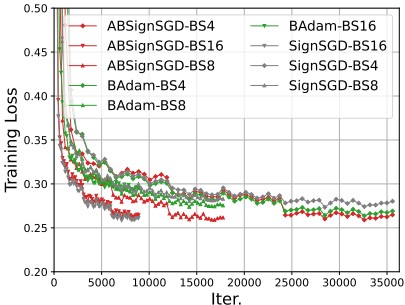

Figure 8: Convergence comparison under decreasing batch sizes (increasing noise). While sign-based methods (ABSignSGD and SignSGD) exhibit higher sensitivity to noise than full-precision baselines (BAdam) as batch size decreases ($16 \rightarrow 4$), ABSignSGD avoids divergence and maintains a performance lead even at the extreme batch size of 4.

### G.2 DISTRIBUTED SCALABILITY (FIXED GLOBAL BATCH)

In the main text (Section 4.3), we demonstrated scalability by fixing the *local* batch size. Here, we present additional scaling results for ABSignSGD-MV in Figure 9, where the global batch size is fixed at 16 while the number of agents varies. Similar to the sensitivity analysis, we use QWEN3-8B on OpenMathInstruct-2 with the fixed optimal baseline learning rate.

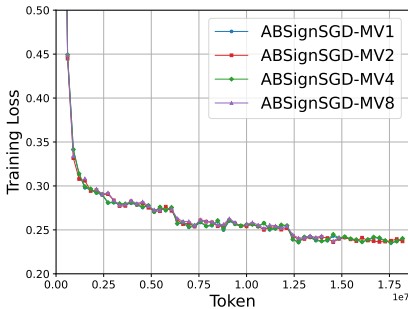

Figure 9: Convergence of ABSignSGD-MV with a **fixed global batch size** of 16. As the number of agents increases, the convergence trajectory remains virtually unchanged, confirming that the Majority Vote aggregation is robust to the number of voters.

### G.3 ROBUSTNESS TO BLOCK SWITCHING SCHEME

We further investigate whether the choice of block update rule impacts the final model quality. Figure 10 compares the downstream accuracy of models trained with different schemes (Depth-Biased, Cyclic, Uniform Random). The results are from finetuning QWEN3-1.7B on OpenMathInstruct-2. Taken together with Figure 3-Right, these results confirm that ABSignSGD is robust to variations in block-switching strategies, thereby offering flexibility and tunability without compromising performance.

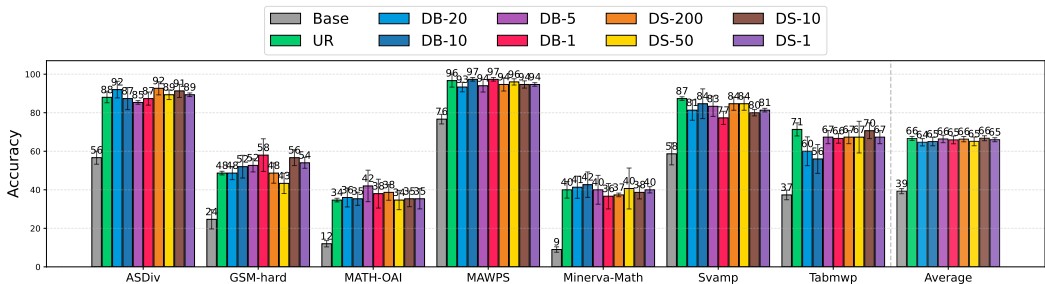

Figure 10: Downstream generalization performance across different block switching schemes. The consistent accuracy confirms that the runtime efficiency gained from the depth-biased update rule does not come at the cost of generalization capability.

## H LLM USAGE

In preparing this manuscript, we made limited use of Large Language Models (LLMs) solely for minor text polishing. Specifically, the LLM was employed to improve grammar, clarity, and readability of certain sentences. All conceptual development, theoretical analysis, experimental design, and result interpretation were conducted entirely by the authors without assistance from LLMs. The scientific content remains the authors' original work.

