# OpenReview forum: "Arbitrary-Order Block SignSGD for Memory-Efficient LLM Fine-Tuning"
_ICLR.cc/2026/Conference — ICLR 2026 Poster_

### Official Review · Reviewer_WMLw · 2025-10-21

**Soundness:** 3
**Presentation:** 2
**Contribution:** 3
**Rating:** 6
**Confidence:** 4

**Summary:**

This paper introduces ABSignSGD, which bridges sign-based optimization and block-coordinate training through an arbitrary-order block update rule and a distributed 1-bit majority-vote aggregation scheme.
The method achieves strong empirical and theoretical performance, offering substantial memory, runtime, and communication efficiency while maintaining competitive accuracy.
Comprehensive ablations and analyses confirm its practical effectiveness for resource-constrained LLM fine-tuning.

**Strengths:**

1. The paper bridges sign-based optimization with block-coordinate methods, introducing an arbitrary-order update rule and 1-bit majority-vote communication that collectively advance memory- and communication-efficient training.

2. Its technical quality is supported by a theoretical convergence analysis, extensive empirical validation on LLMs, and comprehensive ablation and memory analyses that substantiate the design choices.

3. The work offers a simple yet effective solution for large-model fine-tuning under tight hardware constraints.

**Weaknesses:**

1. The method’s reliance on *sign-only updates* discards gradient magnitude information, which may limit precision and hinder convergence in magnitude-sensitive or pretraining scenarios.
2. The approach is primarily evaluated in fine-tuning settings on 8B-scale models, leaving its scalability and effectiveness for full pretraining or larger architectures less explored.
3. The convergence analysis assumes a fixed sign-agreement probability and bounded block-update interval, conditions that may not strictly hold in high-noise or asynchronous environments.
4. While the arbitrary-order rule is conceptually appealing, its implementation details (e.g., latency estimation, update scheduling) could benefit from deeper empirical or theoretical justification.

**Questions:**

Q1. In ABSignSGD, each iteration updates only one active block using the sign of its gradient. Could you clarify whether this *sign-only* update may lead to misalignment between active and inactive blocks across layers during training? How do you ensure stability when different blocks are updated at different frequencies?

Q2. How is a *block* defined in your implementation? Does one block correspond exactly to a Transformer layer, or can it be a smaller unit (e.g., Q, K, V projection matrices) or a larger group of layers? How did you determine the optimal block size (N) in practice? Within a selected block/layer, are all submodules (attention projections, output projection, and feed-forward networks) updated uniformly with the same sign-based rule, or do you apply differentiated treatment among them?


Q3. The paper introduces an “arbitrary-order” block selection strategy with a bounded update interval assumption. Could you elaborate on how the *event-driven* depth-biased rule is implemented in practice? Specifically:
  * How are the latency parameters ( \tau_i ) estimated for different layers?
   * What determines the update readiness timestamp (T_i)?
   * How does this scheduling compare empirically to random or cyclic updates in terms of convergence stability?

Q4. This paper mention that deeper layers are updated more frequently. Could you provide more detail on the precise mathematical rule or heuristic that governs this frequency difference? For example, is it proportional to the inverse of the estimated backward latency or some adaptive function? Since deeper layers are updated more often than shallow ones, does this induce any gradient misalignment or optimization imbalance between early and late layers? Have you observed any degradation in generalization or representation consistency due to this update asymmetry?

Q5. ABSignSGD is designed as a *stateless* and block-switching optimizer. In contrast, widely used adaptive optimizers such as Adam and AdamW maintain two exponential moving averages (the first and second moments), which require continuous accumulation of gradient history for each parameter. Given that block switching interrupts this continuity, is ABSignSGD fundamentally limited to *SGD-like* (stateless) optimizers?
Have you explored—or do you foresee the feasibility of—extending your approach to *stateful* variants that preserve per-parameter moment information while still supporting arbitrary-order block updates?


Q6: The current paper investigates ABSignSGD primarily in the fine-tuning setting, where model parameters start from pretrained weights and the optimization trajectory remains relatively close to the original model. In contrast, pretraining begins from random initialization and requires learning the full parameter distribution from scratch.
Given that ABSignSGD discards gradient magnitude information—updating parameters solely based on gradient signs (±1 per coordinate)—do you expect this sign-only update rule to maintain sufficient precision for large-scale pretraining?
In particular, could the absence of gradient magnitude information hinder convergence when the optimization landscape requires large, magnitude-sensitive adjustments early in training? Have you considered or tested any hybrid variants (e.g., partial sign scaling or adaptive step-size modulation) to mitigate this potential limitation?

Q7: The paper establishes an (O(1\sqrt{K})) convergence rate under mild assumptions using the alignment norm. Could you clarify how sensitive this convergence behavior is to the *sign-agreement probability* (ρ > 0.5) in large-scale, high-noise training regimes?
In particular, do you observe any empirical breakdown when ρ approaches 0.5 (e.g., with small batch sizes or unstable gradients), and how does ABSignSGD behave in comparison to standard SignSGD or BAdam under such conditions?

Q8: For ABSignSGD-MV, you mention a 960× reduction in communication cost compared to standard DDP by transmitting only gradient signs.
Could you elaborate on:

* Whether the MV aggregation is performed per-block or across the full model?
* How sensitive convergence is to the number of participating agents ?
* Whether partial synchronization (e.g., delayed or stale majority votes) degrades convergence stability in multi-node environments?

---

> ### Author Response · Authors · 2025-11-20
>
> Thank you for the detailed assessment and for recognizing the novelty of bridging sign‑based optimization with block‑coordinate training. The constructive feedback regarding scalability, implementation details, and optimization dynamics is greatly appreciated. These points have been addressed below, organized by theme.
>
> ### **1. Scope, Scalability, and Pre-training (Addressing W1, W2, Q6)**
>
> The concern regarding potential information loss in sign-based methods is understandable. However, gradient signs have been shown to carry sufficient information for effective LLM training.  For instance, Chen et al. proposed Lion [1], a pure sign-based optimizer, and demonstrated that it outperforms Adam on specific tasks, including pre-training. This is corroborated by a recent independent research [2], which notes that Lion performs comparably to Adam and suggests that considerations like speed and memory should take precedence in practice. These results collectively show that **gradient-sign information remains highly effective even in large-scale settings**.
>
> The primary focus on fine-tuning (SFT) is a **deliberate design choice** aimed at maximizing utility in resource-constrained environments. Fine-tuning is frequently adopted by practitioners with limited computational capacity, where a memory-efficient and fast algorithm provides the greatest value. The method is intended to help democratize access to LLMs. Although the main results are presented on 8B models—a scale frequently adopted by research and personal use—we also provide additional results fine-tuning **Qwen3-32B** (Figure 5 & 6 in  Appendix E). In these experiments, ABSignSGD again achieves the fastest convergence and best downstream performance with the lowest memory footprint and runtime.
>
> Regarding pre-training, while the method is in principle applicable, the present work focuses on SFT, where memory efficiency is typically prioritized over iteration-wise convergence speed. As shown in our ablation study (Figure 3-Left), ABSignSGD converges more slowly per iteration than Adam. This gap may be influenced by factors such as the absence of momentum or variance adaptation, though their precise impact in combination with sign-based updates remains an open question. Exploring hybrid variants that incorporate such techniques is a potential future direction.
>
> ### **2. On Convergence Assumptions (Addressing W3)**
>
> **Bounded Block-Update Interval:** This condition is strictly satisfied when employing deterministic block-switching schemes. For instance, under our depth-biased update, every block is guaranteed to be selected at least once every $\sum_j \left\lceil \frac{\tau_\max}{\tau_j} \right\rceil$ iterations. Crucially, all our main experiments utilize deterministic updates, ensuring this assumption is satisfied.
>
> *Clarification on Synchronicity:* A potential misunderstanding deserves clarification: the depth-biased update is **fully synchronous**, not asynchronous. Both ABSignSGD and its MV variant wait for the completion of the required blocks before updating. Additional implementation details are provided in the response to Q3.
>
> **Bounded Sign-Agreement Probability:** The manuscript (Line 242) outlines two sufficient conditions for this assumption to hold: (a) the gradient noise variance is element-wise bounded with a sufficiently large mini-batch size, or (b) the gradient noise is unimodal and symmetric. Furthermore, the ablation study provides strong empirical evidence that this condition is satisfied in practice. Even in highly noisy scenarios (Figure 4), only ~1% of parameters have a sign-agreement probability $\rho < 0.5$, while approximately 60% have $\rho > 0.9$. Thus, the theoretical assumption aligns well with practical observation.

---

> ### Author Response · Authors · 2025-11-20
>
> ### **3. On the Arbitrary-Order Rule and Depth-Biased Update (Addressing W4, Q3, Q4)**
>
> Any ambiguity in the manuscript regarding the update mechanics is acknowledged. The update rules are now clarified in **Section 4.1** and **Appendix D.2**, noting that ABSignSGD permits *any* update rule provided the bounded update interval condition is satisfied. The “depth‑biased” update is presented as one such valid choice, prioritizing deeper layers to achieve runtime speedups.
>
> **Clarification on Synchronicity:** A crucial distinction (addressing Q3) is that ABSignSGD and its MV variant are fully **synchronous** algorithms. The previous description (now deleted) that the method "mimics" asynchronous updates was intended as an analogy for the frequency distribution, not the implementation. Each update utilizes fresh block gradients; no gradient staleness or misalignment is introduced.
>
> **Mechanism and Hyperparameters:** To clarify, $\tau$ is a hyperparameter, which does not require estimating the actual hardware latency. The "readiness" time $T_i$ is initialized as $\tau_i$ and updated via $T_{i_k} \leftarrow T_{i_k} + \tau_{i_k}$, where $i_k$ is the chosen block at iteration $k$. As long as $\tau_i$ is smaller for deeper blocks, the intended speedup is achieved.
>
> *Example Trace:* Consider 4 agents where $\tau_i = 7-i$.
>
> | **State**          | **T1 (τ=6)** | **T2 (τ=5)** | **T3 (τ=4)** | **T4 (τ=3)** | **Action**                                 |
> | ------------------ | ------------ | ------------ | ------------ | ------------ | ------------------------------------------ |
> | **Start ($t=0$)**  | 6            | 5            | 4            | **3**        | Agent 4 has min $T$; updates first.        |
> | **Step 1 ($t=3$)** | 6            | 5            | 4            | **6**        | $T_4 \leftarrow 3+3$. Next min is $T_3=4$. |
> | **Step 2 ($t=4$)** | 6            | 5            | **8**        | 6            | $T_3 \leftarrow 4+4$. Next min is $T_2=5$. |
> | **Step 3 ($t=5$)** | 6            | **10**       | 8            | 6            | $T_2 \leftarrow 5+5$.                      |
>
> **Experimental Settings:** In the experiments, $\tau_i$ is set to $N + c(N-i+1)$ with $c=10$. This sets the update frequency of the deepest layer to be approximately $c+1$ times higher than the shallowest layer. This specific $c$ value is chosen to achieve considerable speedup over cyclic updates (e.g., BAdam), rather than to match exact forward/backward ratios.
>
> **Impact on Convergence & Generalization:** As shown in our ablation study (Figure 3-Right), varying the update rule (cyclic, uniform random, depth-biased) has negligible impact on iteration-wise convergence speed or stability. Since the algorithm uses fresh gradients and satisfies the bounded interval assumption, convergence is theoretically guaranteed. Empirically, no generalization degradation is observed; downstream accuracy across different schemes remains stable at $65 \pm 1\%$ (Figure 10). While restricting the update rule could theoretically tighten convergence bounds, it risks reducing flexibility and constraining the discovery of alternative schemes that might yield improvements.
>
> ### **4. Answers to Remaining Questions**
>
> **Q1: Clarification on Misalignment**
>
> The reviewer’s concern about possible misalignment between active and inactive blocks during training is noted. We emphasize that ABSignSGD is strictly **synchronous**: all updates use fresh gradients, so there is no temporal desynchronization. If the concern instead relates to the non-uniform update frequency between active and inactive blocks, the convergence analysis in **Theorem 3.4** is relevant, as it explicitly accounts for varying block update intervals. As long as the maximum update interval is bounded (which is guaranteed by our deterministic schedule), the algorithm converges theoretically. Empirically, the ablation study (Figure 3‑Right) confirms that this frequency variance does not lead to instability or suboptimal optimization trajectories compared to uniform schedules, and Figure 10 further shows that downstream accuracy remains stable across different update schemes.

---

> ### Author Response · Authors · 2025-11-20
>
> **Q2: Block Granularity (N=36)**
>
> In the main experiment with Qwen3-8B, each of the 36 attention layers is defined as a block ($N=36$). This definition is clarified in the revised manuscript (Section 4.1) to ensure reproducibility. While finer or coarser divisions are permissible,  a layer-wise division is chosen as a practical and balanced trade-off. Further division yields diminishing memory returns, as ABSignSGD memory usage scales roughly as $2M + \frac{M}{8N}$. Within each block, all submodules are updated uniformly using the same sign-based rule. A promising future direction is applying block-adaptive learning rates, analogous to methods like Adam-mini [3].
>
> **Q5: Potential Stateful Generalization**
>
> ABSignSGD is not fundamentally limited to stateless optimization. If memory constraints are disregarded, one can maintain global optimizer states and update them block-wise, preserving the runtime speedup of block-coordinate training. However, to achieve **ultra-memory efficiency, runtime speedup, and state accumulation** simultaneously, we propose combining our method with system-level offloading. For instance, global optimizer states can be stored in secondary storage (e.g., CPU memory or NVMe), loading only the required block states when that block is selected. Block-based methods possess a unique advantage here: only the active block's state is required at any given iteration, significantly reducing I/O bandwidth requirements. Consequently, offloading and prefetching of states can be overlapped with the gradient calculation of the current block. By masking this I/O latency with computation, strict memory benefits of block optimization are maintained while achieving the convergence benefits of stateful switching, without incurring significant runtime overhead.
>
> **Q7: Noise Sensitivity**
>
> Additional experiments (Appendix G.1, Figure 8) comparing ABSignSGD, SignSGD, and BAdam across batch sizes from 16 down to 4 highlight the relative noise sensitivity of sign-based methods. These methods show increased sensitivity as batch size decreases, yet no breakdown is observed even at a batch size of 4, where ABSignSGD still converges faster than BAdam. Note that a batch size of 16 is already considered small in standard LLM training, making 4 an extreme lower bound; thus, ABSignSGD maintains its performance advantage in nearly all practical fine-tuning scenarios. The observed sensitivity likely arises from the stochasticity of gradient approximation in the small-batch regimes rather than from the sign update itself. This sensitivity may be mitigated by integrating orthogonal techniques—such as momentum—to stabilize update trajectories, offering a promising direction for future variants.
>
>
>
> **Q8: MV Aggregation and Robustness**
>
> MV aggregation is performed per-block. For $N=30$, this leads to a substantial communication reduction (approx. $960\times$ for 1-bit signs vs 32-bit floats). Additional results confirm the robustness of MV regarding the number of agents:
>
> * **Case A (Fixed Global Batch, Varying Agents):** The global batch is fixed at 16 and vary agents from 1 to 8. Convergence remains virtually unchanged (Appendix G.2, Figure 10).
> * **Case B (Fixed Local Batch, Scaling Agents):** The local batch is fixed at 4 and increase agents from 1 to 32. MV performance tracks the non-MV baseline closely (Section 4.3, Figure 1-Right).
>
> **Regarding asynchronicity**: ABSignSGD-MV is currently strictly synchronous to ensure vote validity. Extending this to an asynchronous setting is non-trivial, as delayed votes would likely introduce staleness and destabilize convergence.
>
> We hope these additional experiments and clarifications address the reviewer's concerns, and are happy to engage in further discussion.
>
> [1] Chen, X., Liang, C., Huang, D., Real, E., Wang, K., et al. "Symbolic Discovery of Optimization Algorithms." *NeurIPS*, 2023.
>
> [2] Zhao, R., Morwani, D., Brandfonbrener, D., Vyas, N., & Kakade, S. "Deconstructing What Makes a Good Optimizer for Autoregressive Language Models." *ICLR*, 2025.
>
> [3] Zhang, Y., Chen, C., Li, N., & He, T. (2024). "Adam-mini: Use Fewer Learning Rates To Gain More." *ICLR, 2025*.

---

### Official Review · Reviewer_UBss · 2025-10-29

**Soundness:** 4
**Presentation:** 2
**Contribution:** 3
**Rating:** 6
**Confidence:** 4

**Summary:**

This paper studies the memory-efficient optimization methods for LLM training. Related works have adopted block-coordinate descent to reduce the memory cost of gradients and optimizer states of the popular optimizers. This paper proposes an algorithm ABSignSGD which introduces SignSGD into the block-coordinate descent to further reduce the memory cost since the signal consumes only 1 bit for one coordinate.

ABSignSGD also adopts a flexible rule of block selection: each block is selected in at most B iterations. The theoretical analysis shows that if the Success Probability Bound is satisfied, ABSignSGD achieves the $O(1/\sqrt{K})$ convergence rate in the form of alignment norm. The experiments of fine-tuning LLMs show that ABSignSGD consumes the least memory and runtime among the baselines. It also converges faster and performs better in downstream tasks than other memory-efficient algorithms. The ablation studies indicate the reason why SignSGD is better than SGD and Adam when coupled with BCD.

The main contribution of this paper is that it goes a step further based on BCD and BADAM by adopting SignSGD as the optimizer in the Block-Coordinate update. The experiment results verify that this is a significant solution for memory-efficient training.

**Strengths:**

SignSGD has been mainly studied in distributed learning or federated learning to reduce the communication cost. On the other hand, recent studies about using Block-Coordinate gradient to reduce the memory cost mainly adopt SGD or Adam as the optimizer. Thus, the originality of this work is good. The authors provide the theoretical analysis for the proposed algorithm and conduct detailed ablation studies to explain why SignSGD performs well in BCD. I think this work is of high quality and significance. Finally, most content of this work is clear and easy to follow.

**Weaknesses:**

This work proposes an extra extension of ABSignSGD, i.e. its distributed version. However, there is no further analysis, including both convergence analysis and experiments for it in the following contents (only a robustness analysis in the supplementary material). The meaning of proposing such an algorithm is not clear.

**Questions:**

This work proposes a “depth-biased” update rule in the experiments, but this part is somewhat confusing. Specifically, ABSignSGD selects the block with the minimal timestamp to update. However, it also says that “This event-driven rule mimics asynchronous execution, where each block becomes eligible for update once its gradient is available”. This seems somewhat contradictory since the former statement indicates that ABSignSGD updates the blocks in a serial way, why it mimics asynchronous execution. In addition, how is the gradient-computation latency obtained?

---

> ### Author Response · Authors · 2025-11-20
>
> We thank the reviewer for the assessment of our work’s soundness and for recognizing the originality and significance of combining SignSGD with Block-Coordinate Descent. Comments on the distributed extension and the update rule are appreciated and addressed below.
>
> ### **1. Analysis of the Distributed (MV) Version (Addressing Weakness)**
>
> We appreciate the reviewer’s request for stronger evidence on the distributed version. While both theoretical analysis and empirical evaluation were already present in the original manuscript, additional clarification and experiments provide a clearer and more complete picture.
>
> - **Theoretical Convergence:** Our convergence analysis in **Theorem 3.4** serves as a **unified framework** for both the standard ABSignSGD and its distributed Majority Vote (MV) variant.
> - **Empirical Evaluation:** The convergence curve for the distributed algorithm was included in **Figure 1-Left** of the original manuscript (labeled *ABSignSGD‑MV*), showing that it closely tracks the single‑node performance. For greater visibility, an **additional plot was added to Figure 1-Right** in the revision.
> - **New Robustness Experiments:** To further validate scalability, experiments were added to test the MV mechanism’s robustness of the MV mechanism with respect to the number of agents:
>   - **Case A (Fixed Global Batch):** The number of agents is varied from 1 to 8 while keeping the global batch size at 16. The convergence curves remain virtually unchanged (Appendix G.2, Figure 10).
>   - **Case B (Fixed Local Batch):** The local batch size is fixed at 4, and the number of agents is increased from 1 to 32 (scaling the global batch size from 4 to 128). The MV performance continues to track the non-MV baseline effectively (Section 4.3, Figure 1-Right).
>
> ### **2. Depth-Biased Update and Latency (Addressing Questions)**
>
> **The description in Section 4.1** has been rewritten to remove the confusing 'asynchronous' analogy. We now explicitly describe the algorithm as synchronous and serial, with $\tau$ defined as a frequency-control hyperparameter rather than a hardware measurement.
>
> - **Clarification on Synchronicity:** ABSignSGD is a **strictly synchronous, serial algorithm**. The term "mimic" was used to illustrate that,  while the algorithm runs serially, the *frequency* at which different blocks are updated varies (deeper layers update more often), similar to how faster nodes might update more frequently in an asynchronous system. However, in implementation, the execution remains serial and deterministic. To avoid confusion, the description on asynchronous update has been deleted.
> - **Latency Estimation:** Critically, the depth-biased update **does not require estimating actual hardware latency**. The "latency" values are simply arbitrary, predefined hyperparameters used to control the update frequency.
> - **Mechanism Example:** Each block is assigned a "readiness" time $T_i$ to each block, initialized as $\tau_i$. The algorithm always selects the block with the minimum $T_i$ to update, then increments that $T_i$ by $\tau_i$. By setting $\tau_i$ to be smaller for deeper layers, they are forced to update more frequently.
>
> **Example Trace:** Consider 4 blocks where we define $\tau_i = 7-i$ (so Block 4 has the smallest $\tau$, representing "fastest" recovery).
>
> | **State**          | **T1 (τ=6)** | **T2 (τ=5)** | **T3 (τ=4)** | **T4 (τ=3)** | **Action**                                 |
> | ------------------ | ------------ | ------------ | ------------ | ------------ | ------------------------------------------ |
> | **Start ($t=0$)**  | 6            | 5            | 4            | **3**        | Block 4 has min $T$; updates first.        |
> | **Step 1 ($t=3$)** | 6            | 5            | 4            | **6**        | $T_4 \leftarrow 3+3$. Next min is $T_3=4$. |
> | **Step 2 ($t=4$)** | 6            | 5            | **8**        | 6            | $T_3 \leftarrow 4+4$. Next min is $T_2=5$. |
> | **Step 3 ($t=5$)** | 6            | **10**       | 8            | 6            | $T_2 \leftarrow 5+5$.                      |
>
> In the main experiments, $\tau_i$ was set to $N + 10(N-i+1)$. This ensures the deepest layers update roughly 11 times more often than the shallowest, but the precise values are flexible hyperparameters, not measurements of hardware latency.

---

### Official Review · Reviewer_qPmE · 2025-10-29

**Soundness:** 3
**Presentation:** 3
**Contribution:** 3
**Rating:** 8
**Confidence:** 4

**Summary:**

The authors propose an algorithm for memory-efficient finetuning of LLMs by combining SignSGD and block-coordinate descent. It has good empirical performance compared to many state-of-art finetuning methods, and the authors also provide convergence proofs for their algorithm.

**Strengths:**

- The algorithm is simple and effective. It combines SignSGD with block-coordinate descent to produce a memory efficient algorithm for finetuning LLMs. It is about 10% more memory efficient than competing methods and also converges faster. It also achieves good generalization performance on many finetuning benchmarks.

- In addition to the base algorithm, the authors also suggest a distributed version that employs majority voting to reduce the communication cost.

- The authors provide convergence proofs of their algorithms, based on previous works from Safaryan & Richtarik 2021.

**Weaknesses:**

- Although the authors propose the communication efficient version of their algorithm using majority voting, I don't see any empirical evaluations on that in the paper.

**Questions:**

- What is the rank of LoRA used in the experiments? How are they selected?

- What are the block size used in the experiments? Are they as small as individual Q,K,V weight matrices or groups of them?

- Where are the results for the block-update rule ablation study at the end of Section 4? I just see the discussion but no tables or figures for that.

---

> ### Author Response · Authors · 2025-11-20
>
> Thanks are extended to the reviewer for their strong support and for highlighting the simplicity, effectiveness, and memory efficiency of the algorithm. The positive assessment is encouraging, and the following clarifications on implementation details are provided.
>
> ### **1. On MV Empirical Evaluations (Addressing Weakness)**
>
> The reviewer’s interest in the distributed setting is appreciated. While the baseline convergence of the MV variant was already included in **Figure 1-Left** (labeled as ABSignSGD‑MV), where it closely tracks the non‑distributed version, we agree that a more robust analysis was needed. Therefore, in the revision an **additional plot in Figure 1-Right** was added for greater visibility, along with **new experiments** testing the robustness of the MV mechanism with respect to the number of agents:
>
> - **Case A (Fixed Global Batch):** The number of agents is varied from 1 to 8 while keeping the global batch size fixed at 16. The convergence curves remain virtually unchanged (Appendix G.2, Figure 10).
>
> - **Case B (Fixed Local Batch):** The local batch size is fixed at 4, and the number of agents is increased from 1 to 32 (scaling the global batch size from 4 to 128). The MV performance continues to track the non-MV baseline effectively (Section 4.3, Figure 1-Right).
>
>
> These results confirm that the majority voting mechanism is robust and scales well.
>
> ### **2. LoRA Rank (Addressing Q1)**
>
> A rank of r=8 was used for all LoRA experiments. This value was chosen because it is the commonly adopted setting in recent fine-tuning literature, including BAdam [1], which used it as a standard baseline.
>
> ### **3. Block Size and Definition (Addressing Q2)**
>
> In the experiments, each attention layer constitutes one block. For the Qwen3-8B model, which has 36 layers, this results in $N=36$ blocks. Section 4.1 has been updated in the revision to explicitly state this definition.
>
> While finer divisions (e.g., treating $W_Q, W_K, W_V$ as separate blocks) or coarser divisions (grouping multiple layers) are permissible under our theoretical framework, layer-wise division is chosen as a practical trade-off. Further dividing the blocks yields diminishing returns on memory efficiency, as the memory usage scales roughly as $2M + \frac{M}{8N}$.
>
> ### **4. Location of Ablation Study (Addressing Q3)**
>
> Apologies for any confusion caused by the layout. The results for the block‑update rule ablation study discussed in Section 4 are presented in **Figure 3-Right**. This figure compares the training loss curves under Cyclic, Uniform Random, and Depth‑Biased update rules, showing that the choice of rule has only a limited impact on iteration-wise convergence speed and stability. Additionally, an **experiment in Section G.3, Figure 10** reports downstream generalization results, demonstrating that accuracy remains stable across different update schemes. Together, Figures 3 and 10 confirm that varying the block‑update rule does not adversely affect either convergence or generalization.
>
> [1] Luo, Q., Yu, H., & Li, X. "BAdam: A Memory Efficient Full Parameter Optimization Method for Large Language Models." *NeurIPS*, 2024.

---

### Official Review · Reviewer_LQY1 · 2025-10-31

**Soundness:** 3
**Presentation:** 3
**Contribution:** 2
**Rating:** 4
**Confidence:** 3

**Summary:**

The paper combines several elements (SignSGD, block-coordinate updates, majority vote) for improving the efficiency of fine tuning LLMs along various dimensions. I suspect the paper is mainly of interest to academic researchers studying optimizer design, compression, or distributed training efficiency. I suspect that few LLM training groups are likely to use sign-based methods in production due to the likely need for substantially more hyperparameter tuning even in cases where they can achieve comparable results to FP-based methods, undoing any efficiency gains.

- The paper consider the fine-tuning step of LLMs.
- Proposes ABSignSGD, combining blockwise updates, 1-bit sign gradients, and arbitrary (depth-biased) scheduling.
- Extends to ABSignSGD-MV for distributed training via 1-bit majority-vote aggregation.
- Provides standard convergence guarantees under sign-agreement assumptions.
- Shows empirical gains in VRAM, runtime, and fine-tuning accuracy vs. LoRA, GaLore, Apollo, and BAdam.
- Lacks validation at larger scales where the claimed advantages would truly matter.

**Strengths:**

- Clear exposition and empirical reproducibility: Algorithms, tables, and ablations are well presented.
- Consistent, modest memory/runtime improvements relative to known baselines at 8B scale.
- Simple implementation concept—no optimizer states, potentially useful for educational or constrained hardware studies.

**Weaknesses:**

- Limited practical relevance: SignSGD and its derivatives are rarely, if ever, used in large-scale LLM tuning, and this is an incremental improvement over SignSGD, making widespread adoption unlikely.
- Limited scope: this paper only addresses fine tuning, a step that takes up only a small part of the compute budget of an LLM.
- Scale mismatch: Experiments are confined to 8B models, but memory and communication constraints dominate only at tens or hundreds of billions of parameters. Small differences on 8B model performance may make the difference between a competitive and uncompetitive large model.
- Questionable net compute advantage: These kinds of methods introduce their own set of hyperparameters, which also require tuning. The paper does not show that the advantages remain when that tuning is taken into account. That is, I may need to conduct quite a few runs of the method in the paper in order to be fairly confident that I have a network that performs as well as straightforward tuning with one of the other methods would have yielded.
- In practice, this would likely be used as just another fine tuning candidate, and people would use its output only if it happens to yield better accuracy; in effect, a method intended for generating for improving compute performance would end up being just another ad hoc fine tuning variant.

**Questions:**

- Can you respond about the concerns about hyperparameter tuning and the overall process efficiency?
- Why did you not try to fine-tune larger LLMs? It seems to me this paper would be much stronger if you could directly demonstrate that the method allows some fine tuning of large models on limited hardware that are impossible to fine tune using any of the other methods.

---

> ### Author Response · Authors · 2025-11-20
>
> We sincerely thank the reviewer for their detailed feedback. The reviewer's central concerns appear to stem from two main points: 1) a belief that sign-based methods are impractical for production LLM training due to "questionable net compute advantage" from "substantially harder hyperparameter tuning", and 2) a concern about the "limited scope" (fine-tuning only) and scale (8B models) of our experiments.
>
> The following clarifications address these points.
>
> ### **1. On Practical Adoption, Tuning Difficulty, and Net Compute Advantage**
>
> We acknowledge the reviewer's point that AdamW is the dominant optimizer in production. For large-scale use, any new optimizer must offer a *substantially* better performance/cost trade-off to overcome the "shifting cost" of replacing a well-established and deeply integrated tool. The reviewer suggests this high cost is due to tuning, but our results indicate that sign‑based methods are not "substantially harder to tune".
>
> - **Tuning Difficulty:** In fact, most popular sign-based methods have **fewer** hyperparameters than AdamW. For instance, SignSGD (learning rate $\alpha$) and Signum ($\alpha, \beta$) are simple. A prominent recent example is Lion [1] ($\alpha, \beta_1, \beta_2$), a sign-based optimizer specifically developed for training large models that has gained significant community adoption and has been shown in multiple papers to be a strong competitor to Adam. These methods require tuning at most three parameters, whereas AdamW requires four ($\alpha, \beta_1, \beta_2, \epsilon$).
>
> - **Independent Validation:** Our position is supported by a recent, rigorous independent study [2]. This work found that Lion and Signum all achieve similar performance and robustness to hyperparameters compared to Adam. Their key conclusion is that considerations like speed and memory should take precedence—which is precisely the focus and core contribution of our paper.
>
> - **ABSignSGD Tuning and Net Compute Advantage:** The reviewer correctly notes that our block-wise approach *could* introduce new hyperparameters (from block partition and switching). However, our ablation study (Sec 4.5, Fig 3-Right) confirms that convergence is not sensitive to most block-switching rules (with additional results in Appendix G.3, Figure 10 implying generalization is not affected either). **In fact, a standard partition (e.g., by transformer layer) and the depth-biased rule with a fixed hyperparameter already yield superior performance across different model sizes, architectures, and tasks.**
>
>   In practice, this means ABSignSGD effectively has only one major hyperparameter, the learning rate, to tune (three less than Adam). Moreover, the empirical study showed **no sensitive response from ABSignSGD to its learning rate either.** This simplicity creates a **clear net compute advantage** when considering the entire fine-tuning process. **All methods were tuned via a grid search on the learning rate over an identical logarithmic grid size.** The efficiency of our method is highlighted by hardware allocation: **one GPU was used to run and tune ABSignSGD and BAdam, one GPU for LoRA, one for Apollo, and two for GaLore.**  This demonstrates that the end-to-end process for ABSignSGD is significantly more efficient, even when accounting for tuning.
>
> Given this evidence, sign-based methods are not inherently harder to tune, and their significant speed and memory advantages present a compelling case for their adoption.
>
> ### **2. On the "Limited Scope" of Fine-Tuning**
>
> The reviewer is concerned that only fine-tuning is addressed,  which they view as a "small part of the compute budget."
>
> However, fine-tuning is far from a "limited case" and that our focus on it was a deliberate research choice. While pretraining consumes a massive one-time budget, fine-tuning is performed far more frequently by a much wider range of users, from researchers to small companies.
>
> It is in this resource-constrained setting—where users do not have access to large-scale pretraining clusters—that speed and memory efficiency matter most. Therefore, improving the efficiency of fine-tuning is critical for democratizing access to LLMs, making it a highly practical and impactful problem.

---

> ### Author Response · Authors · 2025-11-20
>
> ### **3. On the Scale of Experiments**
>
> The reviewer observes that our main experiments were on 8B models and questions whether this scale is representative.
>
> - **New 32B Experiments:** To address this concern, **the manuscript is updated to include** new experimental results on a 32B model (**see Appendix E, Figures 5 & 6**).  These new results are consistent with our findings on 8B models, demonstrating that the benefits of ABSignSGD hold at a larger scale.
> - **Justification for 8B Focus:** The initial focus on 8B models was based on two factors: 1) the limited computing resources available to us, and 2) the fact that models of this scale represent a very common and practical use case for full-parameter fine-tuning (as comparatively few users attempt to fine-tune models with hundreds of billions of parameters).
>
> ### **Conclusion**
>
> Sign-based methods demonstrate significant potential for LLM training, especially in resource-limited settings. The proposed ABSignSGD consistently delivers superior performance under smaller budgets. When considering the *entire* process—including a tuning effort that is no more difficult (and arguably simpler) than AdamW—ABSignSGD provides a clear net compute advantage, as evidenced by the experimental setup. We hope this response along with the additional experimental results clarifies our contributions and addresses the reviewer's valid concerns.
>
> [1] Chen, X., Liang, C., Huang, D., Real, E., Wang, K., et al. "Symbolic Discovery of Optimization Algorithms." *NeurIPS*, 2023.
>
> [2] Zhao, R., Morwani, D., Brandfonbrener, D., Vyas, N., & Kakade, S. "Deconstructing What Makes a Good Optimizer for Autoregressive Language Models." *ICLR*, 2025.

---

> ### Author Response · Authors · 2025-11-26
> **Gentle Follow-up: New 32B Experiments and Practicality Analysis**
>
> Dear Reviewer LQY1,
>
> Thank you again for your constructive review.
>
> As the discussion period is coming to a close, we wanted to gently follow up to ensure you have had a chance to see our response and the revised manuscript. We took your concerns regarding **scalability** and **practical adoption** very seriously and have updated the paper to address them:
>
> - **Scalability:** We conducted new fine-tuning experiments on **Qwen3-32B** (Appendix E). The results confirm that ABSignSGD maintains its efficiency and performance advantages at this larger scale, addressing your concern regarding the 8B limit.
> - **Practicality & Tuning:** We explicitly added a discussion on the **Lion optimizer** to highlight the growing community adoption of sign-based methods. Furthermore, we added an ablation study on block-switching schemes (Figure 3-Right) to demonstrate that ABSignSGD is robust and does not introduce complex tuning overhead.
> - **Scope (Fine-Tuning):** We also expanded our discussion on why we prioritize Fine-Tuning (SFT). While we acknowledge your point on pre-training, our focus is on democratizing access for resource-constrained practitioners, where SFT is the dominant workload.
>
> We would be very grateful for your thoughts on these new results and are happy to answer any further questions.
>
> Best regards,
>
> The Authors

---

### Author Response · Authors · 2025-11-26
**General Response: Summary of New Experiments (32B Scaling, Robustness) and Clarifications**

Dear Area Chair,

Thank you and all the reviewers for your time and the valuable feedback on our submission, "Arbitrary-Order Block SignSGD for Memory-Efficient LLM Fine-Tuning" (Submission 10064). We have carefully considered all comments and have prepared a detailed rebuttal.

While we have addressed each point specifically in our individual rebuttals by guiding the reviewers to the exact locations in the revised manuscript, we feel it is important to summarize these key instances for your convenience, as they are central to the evaluation of our work’s contribution and completeness.

### 1. Validating Scalability: New 32B Model Experiments

**Reviewers LQY1** and **WMLw** questioned whether the efficiency gains observed on 8B models would hold at larger scales. To address this, we have added comprehensive experiments fine-tuning **Qwen3-32B** (approx. 4x larger than the original main experiments). As detailed in **Appendix E (Figures 5 & 6)**, ABSignSGD consistently maintains its advantages at this scale, achieving the fastest iteration-wise and wall-clock convergence, along with superior downstream accuracy compared to other baselines. This confirms the method scales effectively to larger architectures.

### 2. Practicality & Net Compute Advantage

**Reviewer LQY1** raised points regarding the "net compute advantage," suggesting that sign-based methods might be difficult to tune. We address this in two parts:

- **A. Community Adoption:** We highlight that sign-based methods have gained significant traction in the community, most notably with the **Lion optimizer** [1]. Independent studies (e.g., [2]) have confirmed that sign-based optimizers achieve similar performance and robustness to hyperparameters compared to AdamW.
- **B. Ease of Tuning:** Unlike AdamW (4 hyperparameters) or Lion (3), **ABSignSGD effectively has only one major hyperparameter to tune: the learning rate.** Additional ablation study in **Appendix G.3** and prior results in **Figure 3-Right** show that convergence and downstream performance are insensitive to the specific block schedule, indicating that practitioners do not need to tune the related hyperparameters to achieve competitive results.

### 3. Distributed Robustness: Majority Vote (MV)

**Reviewers qPmE** and **UBss** noted an absence of empirical evidence regarding the performance of the ABSignSGD-MV extension. While baseline results were included in the original submission (**Figure 1-Left**), we recognized the need for greater visibility and expanded the evaluation. Varying agents from 1 to 8 (keeping global batch fixed) showed virtually unchanged convergence (**Appendix G.2, Figure 9**). Furthermore, scaling agents from 1 to 32 (scaling global batch up to 128) showed that MV tracks the single-node baseline closely (**Figure 1-Right**).

### 4. Noise Sensitivity & Stability

**Reviewer WMLw** raised questions about the potential breakdown of sign-based updates in high-noise regimes. We conducted stress tests reducing the batch size from 16 down to 4 to simulate extreme noise. As shown in **Figure 8 (Appendix G.1)**, while convergence slows, ABSignSGD **does not diverge** and maintains a performance lead over BAdam even at the extreme batch size of 4.

### 5. Clarification on "Depth-Biased" Update

**Reviewer UBss** and **WMLw** found the description of the depth-biased update rule (referencing "latency" and "mimicking asynchronous execution") confusing. We rewrote **Section 4.1** and **Appendix D.2** to clarify that ABSignSGD is a strictly **synchronous** algorithm. We clarified that the "latency" parameters ($\tau$) are simply user-defined hyperparameters to control update frequency, not measurements of actual hardware latency.

------

**Conclusion** We have updated the manuscript to incorporate these findings and enhance clarity. We believe these new results strengthen the paper's contribution and address the gaps noted by reviewers. Thank you for considering this additional context during your final assessment.

Thank you for your time and consideration.

Best regards,

The Authors of Submission 10064

---

### Meta-Review · Area_Chair_huhA · 2026-01-06

**Summary:**

This paper proposes ABSignSGD, a memory-efficient optimizer for fine-tuning LLMs that combines block-coordinate descent with sign-based updates and a flexible scheduling rule. It demonstrates superior memory savings and faster wall-clock convergence compared to baselines like LoRA and BAdam on models up to 32B parameters.

Reviewers identified many strengths of the proposed method. Despite that, the most crucial concerns are listed below:
1. some reviewers are worried that sign-based methods are not used in production and that 8B models were too small to prove the method's worth
2.  There was concern that block-wise updates would make tuning too difficult
3.  Reviewers were confused if the method was asynchronous and how "latency" was measured
4.  Some reviewers questioned if discarding gradient magnitude would hurt convergence or stability in noisy settings
5.  some reviewers noted a lack of empirical evidence for the Majority Vote (MV) extension in the initial draft

**Reviewer Concerns:**

# Addressed Concerns
The rebuttal was quite detailed and I feel it addressed some concerns that reviewers had, including:
*  issue related to scalability to larger models, as originally, the paper only used 8B models. The authors added experiments for Qwen3-32B, showing consistent gains
* some reviewers feared the block-coordinate approach added too many parameters to tune. The authors proved that the block schedule does not need tuning to be effective and that the method is robust with just the learning rate
*  some reviewers were confused by the "asynchronous" terminology and how latency was measured. The authors clarified the method is strictly synchronous and latency is just a user-defined control for update frequency
*  Reviewers noted a lack of evidence for the Majority Vote (MV) extension.  New results in Appendix G.2 and Figure 1-Right show the MV variant tracks the single-node performance reliably up to 32 agents
*  There was concern that sign-only updates would break down with noisy gradients. Stress tests with an extreme batch size of 4 showed the method is stable and still beats the BAdam baseline

# Outstanding Concerns
*  While the rebuttal addresses fine-tuning well, some concern about whether sign-only updates provide enough precision for full pre-training (from scratch) remains mostly theoretical. The authors acknowledge this is out of scope for the current paper
*  there was a fear that the method is limited to stateless (SGD-like) optimizers. The authors suggested a system-level offloading fix for the future, but a practical, integrated "stateful" version of the algorithm is not yet demonstrated in this work

**Reviewer Scores:**

*  Reviewer LQY1, Original Score: 4. This reviewer main problem was "scale mismatch" and if method is practical. Since authors added 32B model results that show same advantage, I feel they would most likely increase their score.
*  Reviewer qPmE, Original Score: 8. This reviewer was already very happy and I feel he would keep his score.
*  Reviewer UBss, Original Score: 6 . This reviewer liked the originality but was confused by the "depth-biased" rule and the lack of distributed analysis. During the rebuttal, I feel that authors clarified the issues and he would most likely keep or increase his score.
*  Reviewer WMLw, Original Score: 6.  I feel that the rebuttal could address his concerns in which case he could increase the score, but there is also a good chance he could keep his current one.

---

### Decision · Program_Chairs · 2026-01-26

Accept (Poster)